# Multiscale mechanisms of green concrete regulated by silicon-to-calcium ratio: Physico-mechanical properties, hydration structure, and durability performance

Jiayuan Lou[1], Yiwen Liang[1]*, Wenhua Zha[2], Qiang Su[1]

**1** College of Civil Engineering and Architecture, Quzhou University, Quzhou, Zhejiang, China, **2** School of Civil and Architectural Engineering, East China University of Technology, Nanchang, Jiangxi, China

* 38102@qzc.edu.cn

## Abstract

To explore efficient pathways for the resource utilization of silicon-rich solid wastes in low-carbon concrete, this study proposes a synergistic regulation strategy centered on the silicon-to-calcium (Si/Ca) ratio. Three types of silicon-rich solid wastes-glass sand, glass powder, and rice husk ash-were incorporated to produce waste glass and rice husk concrete (WGRC). The effects of varying Si/Ca ratios on the workability, mechanical properties, and durability of WGRC were systematically investigated. Furthermore, the underlying mechanisms were elucidated through microstructural analysis. The results indicate that WGRC exhibits optimal overall strength within a Si/Ca ratio range of 0.46~0.58. When the Si/Ca ratio ranged from 0.52~0.58, WGRC demonstrated superior resistance to water penetration and sulfate attack, with the lowest mass loss rate (0.54% after 180 drying-wetting cycles) and the smallest ultrasonic velocity reduction (only 2.6%). At a Si/Ca ratio of 0.58, the carbonation resistance was maximized, yielding the lowest carbonation rate. In addition, the Si/Ca ratio within the C-S-H gel increased progressively with curing age, though the rate of increase slowed after 90 days. A shear damage constitutive model was developed to accurately describe the nonlinear response characteristics under varying Si/Ca ratios and shear angles, validating the coupling relationship among composition, structure, and performance. These findings provide new theoretical insights and design strategies for the synergistic utilization of multiple solid wastes in low-carbon concrete. They also offer a scientific basis for enhancing the mechanical and durability performance of WGRC, thereby contributing significantly to the advancement of sustainable construction materials.

**Data availability statement:** All relevant data are within the paper.

**Funding:** This research was supported by the 2024 National College Student Innovation and Practice Program (Grant No.: 202411488054).

**Competing interests:** The authors have declared that no competing interests exist.

## 1. Introduction

According to statistics, approximately 14 billion tons of concrete are consumed worldwide each year, resulting in nearly 2.5 billion tons of $CO_2$ emissions, with cement production accounting for more than 90% of the total carbon emissions over the entire life cycle of concrete [1]. It is important to note that this environmental cost and resource consumption contradiction is particularly acute in developing countries [2]. In China, cement production reached 2.1 billion tons in 2022, accounting for 55% of the global total; however, $CO_2$ emissions per ton of cement were as high as 0.82 tons, significantly exceeding the international advanced level of 0.65 tons. At the same time, the accumulation of urban and agricultural waste continues to worsen. More than 130 million tons of waste glass are generated globally each year, with a recycling rate of less than 30%. Rice husk ash, a by-product of rice processing, has an annual output of 150 million tons, but more than 90% is directly discarded or land-filled, leading to land acidification and PM2.5 pollution [3,4]. Against this background, considering the transformation of waste glass and rice husk ash into alternative materials for concrete to achieve the coordinated development of "low-carbon cement and solid waste resource utilization" has become a crucial research direction in the field of concrete materials. This approach not only reduces cement consumption and carbon emissions but also alleviates the pressure of solid waste disposal, aligning with the development goals of the circular economy and low-carbon buildings [5–8].

Recently, grinding waste glass into different particle sizes to obtain waste glass sand or waste glass powder, using waste glass sand to replace fine aggregates, and using waste glass powder to replace cement have provided new directions for optimizing concrete performance [9–11]. Omoding et al. [12] reported that replacing fine aggregates with waste glass sand can improve aggregate grading in concrete, and the low water absorption and high hardness of waste glass sand enhance the wear resistance of concrete. Additionally, previous studies have indicated that the fluidity of waste glass sand concrete is higher than that of ordinary concrete [13–16]. Research on waste glass powder concrete has shown that replacing cement with an appropriate amount of waste glass powder can achieve 90% of the compressive strength of the control group at 28 days, although early strength loss can reach 20% to 30% [17–20]. However, when the content of waste glass powder exceeds 30%, insufficient cementitious activity may lead to a sharp decline in strength [18,19]. It is worth noting that, unlike waste glass sand and waste glass powder, the application of rice husk ash presents another dimension. A key feature is that rice husk ash contains more than 90% $SiO_2$, and its nano-scale particle size effectively fills the internal pores of concrete while promoting the densification of calcium-silicate-hydrate gel through the pozzolanic reaction [20]. However, when rice husk ash is used alone as a cement substitute, the calcium-silicon ratio in the cementitious system may drop below 1.0, resulting in a 2~3 times increase in setting time and a significant reduction in early strength [21].

It is well known that the calcium-silicon ratio is a key chemical parameter of calcium-silicate-hydrate gel, directly affecting the mechanical properties and durability of concrete [22,23]. In ordinary Portland cement, the calcium-silicon ratio typically ranges

from 2.0 to 3.5, and the hydration products generally exhibit a layered or fibrous structure, providing high early strength but often leading to a higher porosity [24,25]. When silicon-rich materials are introduced into concrete, the calcium-silicon ratio significantly decreases, and hydration products gradually transform into low-calcium types (similar to a tobermorite-like structure). This increases the degree of polymerization, refines the microstructure, and improves the later strength and impermeability of concrete [26–28]. Notably, the chemical characteristics of waste glass powder and rice husk ash are complementary: the calcium oxide in waste glass helps mitigate the decline in the calcium-silicon ratio, while the high silicon dioxide content in rice husk ash accelerates calcium ion consumption. This synergistic effect may dynamically regulate the calcium-silicon ratio in the cementitious system, optimizing the structure and performance of hydration products. However, given the high $SiO_2$ content in silicon-rich materials, the authors propose that adopting a reverse-thinking approach by studying the silicon-calcium ratio can provide a better understanding of the hydration reaction mechanism of these materials.

Based on this, this study proposes a "multi-solid waste collaborative design approach with the silicon-calcium ratio as the core regulation parameter." The goal is to use the combined system of waste glass sand, waste glass powder, and rice husk ash to optimize aggregate physical properties through waste glass sand, supply calcium elements through cement, and provide silicon elements and pore-filling effects through waste glass powder and rice husk ash, thereby achieving both resource recycling and performance enhancement in concrete. Eight concrete mix proportions were designed to systematically evaluate physical and mechanical properties as well as durability. The grey relational analysis method was used for principal component evaluation, energy dispersive X-ray spectroscopy was employed to measure the surface elemental content of calcium-silicate-hydrate gel, and Microstructural morphology was characterized using a scanning electron microscope. This study focuses on addressing the following key questions: (1) How does the silicon-calcium ratio influence the structure of calcium-silicate-hydrate gel in multi-solid waste systems? (2) What are the effects of the silicon-calcium ratio on concrete performance when multiple solid wastes are used as replacements? The scientific value of this study lies in filling the gap in the "composition-structure-performance" correlation model under multi-solid waste collaboration, providing a reference for designing low-carbon concrete from the perspective of the silicon-calcium ratio. The technical roadmap of this study is shown in Fig 1.

## 2. Materials and research methodology

### 2.1 Materials

The raw materials used in this study are as follows:

(1) Cement: P·O 42.5 ordinary portland cement with a density of $3.15\,g/cm^3$, a specific surface area of $350\,m^2/kg$, and a 28-day compressive strength of 42.5 MPa.

(2) Fine aggregate: natural river sand with a fineness modulus of 2.8, a mud content of approximately 0.8%, and a density of $2.65\,g/cm^3$.

(3) Coarse aggregate: continuously graded crushed stone with a particle size of $5\sim10\,mm$ and a crushing value of 8.2%.

(4) Waste glass sand: obtained by recycling and grinding waste glass, with a particle size of $0.3\sim0.6\,mm$.

(5) Waste glass powder: obtained by recycling and grinding waste glass, with a particle size of approximately $40\,\mu m$.

(6) Rice husk ash: collected from biomass power plants, with a specific surface area of $1200\,m^2/kg$.

The chemical compositions of cement, waste glass, and rice husk ash were determined through X-ray fluorescence spectroscopy analysis, as shown in Table 1.

### 2.2 Mix proportions

Based on the silicon-to-calcium ratio (Si/Ca ratio) regulation design concept, eight groups of samples were designed, with the Si/Ca ratio starting at 0.34 and increasing in increments of 0.06 up to 0.76. It should be noted that in the basic mix

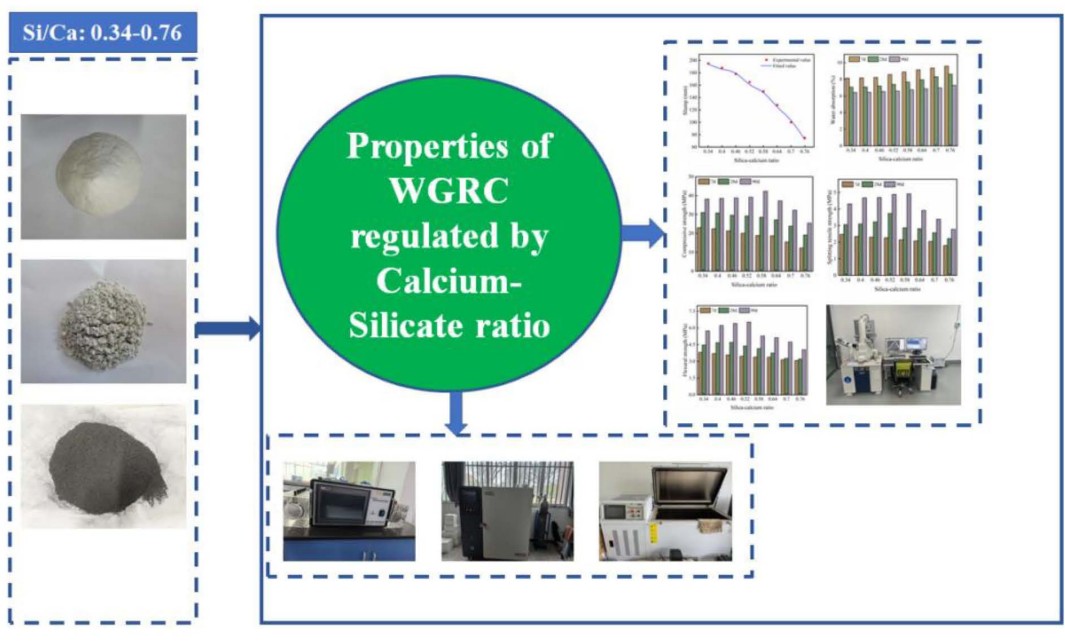

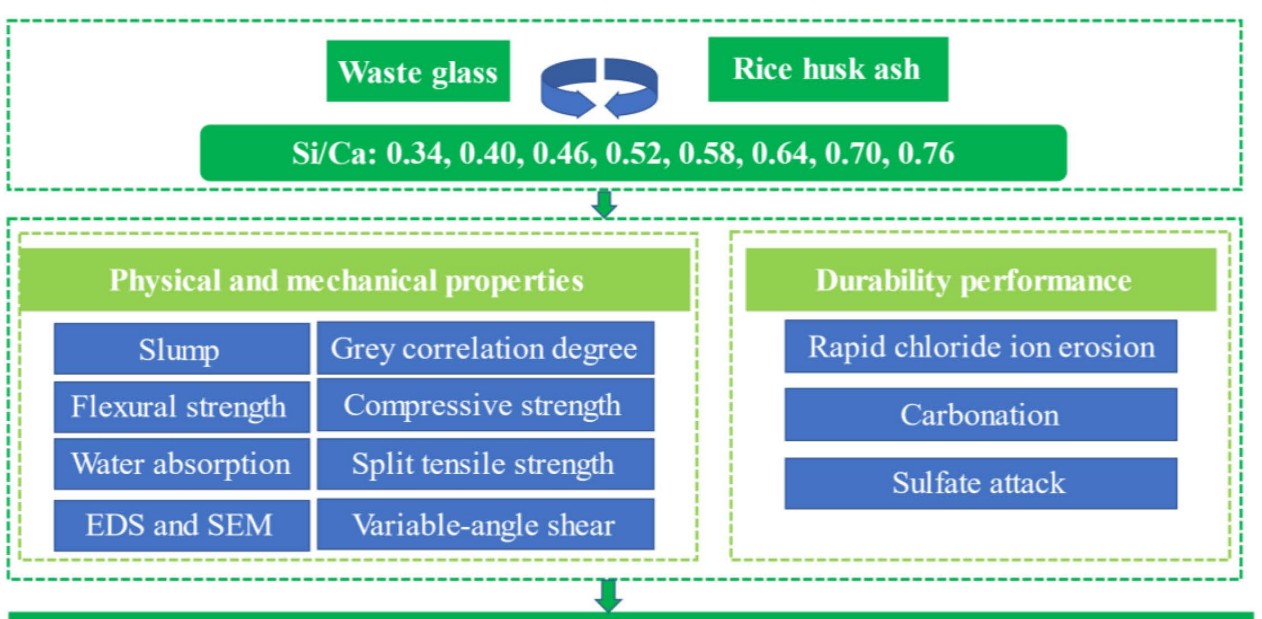

**Fig 1. Technology roadmap.**

proportion design, the cement dosage was set at 368 kg/m³, and waste glass sand was used to replace 30% of the fine aggregate. The amounts of fine aggregate, coarse aggregate, and waste glass sand were 443.1 kg/m³, 1202 kg/m³, and 189.9 kg/m³, respectively. The Si/Ca ratio was calculated based on the molar ratio of $SiO_2$ to CaO derived from the oxide

**Table 1. Chemical composition of cement, glass sand and rice husk ash (Unit: %).**

| Type | SiO$_2$ | Fe$_2$O$_3$ | CaO | Al$_2$O$_3$ | K$_2$O | MgO | Na$_2$O | SO$_3$ | TiO$_2$ |
|------|---------|-------------|-----|-------------|--------|-----|---------|--------|---------|
| Cement | 20.6 | 3.86 | 63.14 | 5.59 | 1.33 | 1.65 | 0.137 | 3.09 | 0.289 |
| Glass | 68.52 | 0.378 | 11.53 | 2.587 | 0.549 | 1.31 | 14.23 | 0.213 | 0.0579 |
| Rice husk ash | 92.30 | 0.179 | 1.14 | 0.256 | 3.88 | 0.723 | 0.041 | 0.224 | 0.031 |

compositions of the raw materials listed in Table 1. Specifically, the molar amounts of SiO$_2$ and CaO contributed by each binder component were determined by dividing their mass fractions by the corresponding molar masses. Referring to the chemical compositions in Table 1, the intrinsic Si/Ca ratios of cement, GP, and RHA were calculated as 0.304, 5.54, and 76.8, respectively. In this study, the dosage ratio of GP to RHA was fixed at 1:1 to maintain a balanced contribution of silica sources and to control variables in the mixture design, and their total contents were adjusted to achieve the target Si/Ca ratios. The specific mix proportions and corresponding Si/Ca ratios are presented in Table 2.

## 2.3 Experimental scheme

The experiments to be conducted in this study include slump, water absorption, compressive strength, splitting tensile strength, flexural strength, Energy Dispersive X-ray Spectroscopy (EDS), Scanning Electron Microscope (SEM), rapid chloride ion penetration, carbonation, and sulfate erosion. The specific experimental scheme is as follows:

(1) Slump test will be conducted during the preparation of concrete samples. The fresh mixture will be layered into the slump cone and compacted until full, then the cone will be lifted vertically, and the slump height will be measured [29].

(2) Water absorption will be tested at 7, 28, and 90 days of curing. The test will be carried out using the boiling water method: First, the dried specimens will be placed in a water tank, with water added to fully submerge the specimens. The water will be heated to boiling and maintained for 5 hours. The specimens will then be naturally cooled in water, removed, and dried to remove surface water. The weight will be recorded, and the water absorption rate will be calculated [30,31].

(3) Compressive strength, splitting tensile strength, and flexural strength will be tested at 7, 28, and 90 days of curing. All tests will be conducted with displacement-controlled loading. The loading rate for compressive strength will be 3 mm/min, for splitting tensile strength will be 1.5 mm/min, and for flexural strength will be 1.0 mm/min [32].

(4) Variable-angle shear tests were conducted at 28 days of curing. The shear angles were set at 25°, 45°, and 65°, respectively. The selected shear angles (25°, 45°, and 65°) were chosen to represent low, intermediate, and high inclination conditions, with 45° corresponding to a commonly adopted reference angle in shear-related studies. The specimens were cubic in shape with dimensions of 50 mm × 50 mm × 50 mm. The tests were carried out under displacement-controlled loading at a constant rate of 0.5 mm/min until specimen failure.

(5) EDS and SEM tests were performed using a Hitachi S-4800 microscope (as shown in Fig 2), with a resolution of 1.2 nm, a magnification of 2 kx, and an accelerating voltage of 15 kV. To ensure the accuracy and reliability of the EDS and SEM results, a ~10 nm gold coating was applied to the specimen surfaces using a vacuum sputter coater prior to testing, in order to enhance surface conductivity [33,34].

(6) Rapid chloride ion erosion: concrete samples will be cut to standard sizes (100 mm in diameter and 50 mm in height) and saturated using a vacuum saturation device. The TYC-RCMPN concrete chloride ion permeability comprehensive measuring instrument (as shown in Fig 3) will be used, with 5.0% NaCl solution and deionized water added to the two

**Table 2. Mix ratio (Unit: kg/m³).**

| Sample | Silica-calcium ratio | Cement | Fine aggregate | Coarse aggregate | Glass sand | Glass powder | Rice husk ash | Water |
|---|---|---|---|---|---|---|---|---|
| S/C-0.34 | 0.34 | 357.06 | 443.1 | 1202 | 189.9 | 5.47 | 5.47 | 208 |
| S/C-0.40 | 0.40 | 346.12 | 443.1 | 1202 | 189.9 | 10.94 | 10.94 | 208 |
| S/C-0.46 | 0.46 | 335.18 | 443.1 | 1202 | 189.9 | 16.41 | 16.41 | 208 |
| S/C-0.52 | 0.52 | 324.24 | 443.1 | 1202 | 189.9 | 21.88 | 21.88 | 208 |
| S/C-0.58 | 0.58 | 313.30 | 443.1 | 1202 | 189.9 | 27.35 | 27.35 | 208 |
| S/C-0.64 | 0.64 | 302.36 | 443.1 | 1202 | 189.9 | 32.82 | 32.82 | 208 |
| S/C-0.70 | 0.70 | 291.42 | 443.1 | 1202 | 189.9 | 38.29 | 38.29 | 208 |
| S/C-0.76 | 0.76 | 280.48 | 443.1 | 1202 | 189.9 | 43.76 | 43.76 | 208 |

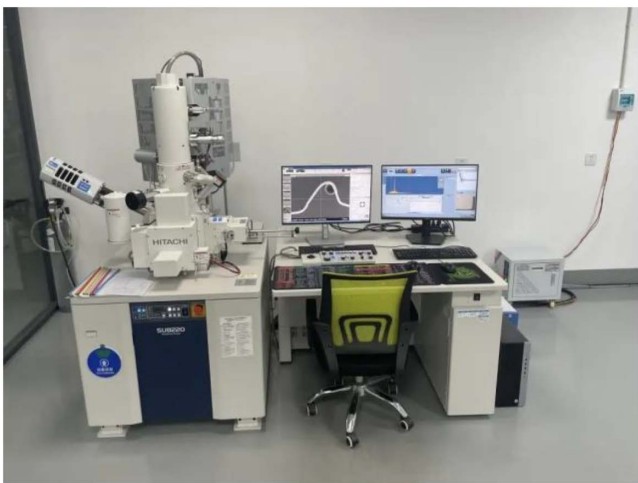

**Fig 2. EDS experimental instrument.**

sides of the instrument. The experiment will be conducted for 6 hours, and the electrical flux data during the experiment will be recorded.

(7) Carbonation test: after curing for 28 days under standard conditions, the samples will be placed in a carbonation chamber (shown in Fig 4) with $CO_2$ concentration controlled at 20%, temperature at 20 °C, and humidity at 70%. Samples will be taken at carbonation times of 3, 7, 14, and 28 days for carbonation depth and pH value testing.

(8) Sulfate erosion test: after curing for 28 days under standard conditions, the samples will be fully immersed in a 5% sodium sulfate solution for 16 hours. The samples will then be removed, dried at 60 °C for 6 hours, and cooled at room temperature for 2 hours. After 30, 60, 90, 120, 150, and 180 wet-dry cycles, mass and ultrasonic velocity tests will be conducted. The sulfate wet-dry cycle experimental machine is shown in Fig 5.

## 3. Physical and mechanical properties

### 3.1 Slump

The slump of waste glass and rice husk ash concrete (WGRC) with different silicon-calcium ratios is shown in Table 3. It was observed that as the silicon-calcium ratio increased, the slump of WGRC decreased. This may be because cement,

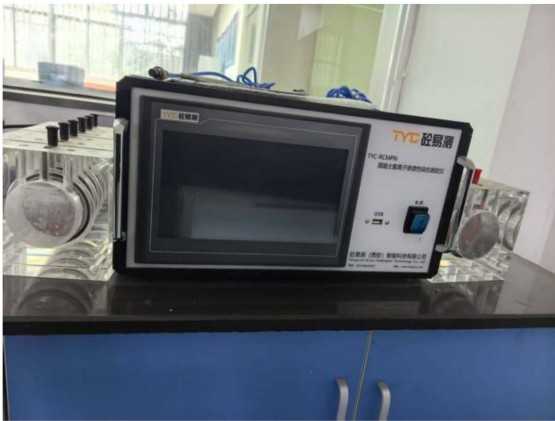

**Fig 3. Chloride ion erosion tester.**

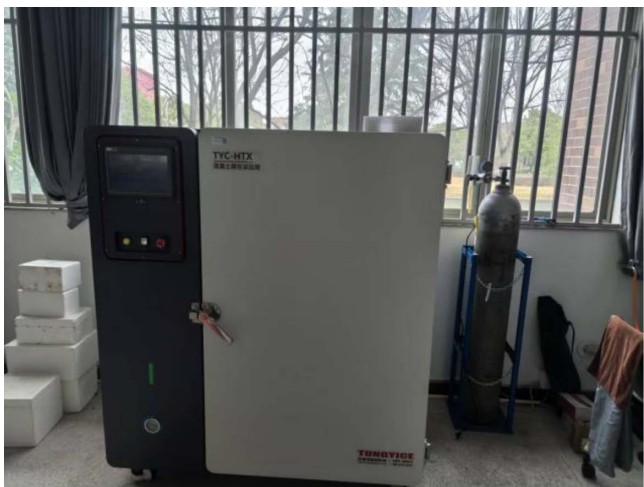

**Fig 4. Carbonization box.**

glass powder, and rice husk ash formed other calcium hydration products during the mixing process. The microstructure of these products may increase the viscosity of the cement paste, thus reducing the flowability of the concrete [35]. It should be noted that the decrease in slump of WGRC exhibited a staged characteristic: when the silicon-calcium ratio increased from 0.34 to 0.58, the slump decreased from 195 mm to 150 mm, a reduction of 45 mm. On average, for every 0.06 increase in the silicon-calcium ratio, the slump decreased by about 10~15 mm; when the silicon-calcium ratio increased from 0.58 to 0.76, the slump decreased from 150 mm to 75 mm, a total reduction of 75 mm. On average, for every 0.06 increase in the silicon-calcium ratio, the slump decreased by about 22~28 mm. This is primarily because, when the silicon-calcium ratio is less than 0.58, the incorporation of glass powder and rice husk ash mainly serves to fill microvoids, optimizing the paste structure and reducing the confinement of free water, which allows the flowability to remain at a relatively high level.

However, when the silicon-calcium ratio exceeds 0.58, the $SiO_2$ content significantly increases, promoting the formation of a large amount of C-S-H gel, which quickly thickens the paste, thus exacerbating the loss of flowability and leading to

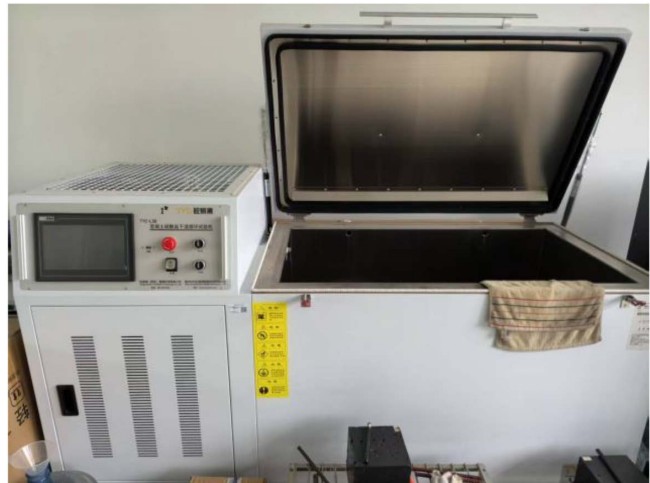

**Fig 5. Sulfate dry and wet cycle test machine.**

**Table 3. Slump test results.**

| Silica-calcium ratio | 0.34 | 0.40 | 0.46 | 0.52 | 0.58 | 0.64 | 0.70 | 0.76 |
|---|---|---|---|---|---|---|---|---|
| Slump/mm | 195 | 188 | 178 | 165 | 150 | 128 | 100 | 75 |
| Reduction range/% | _ | 3.59 | 8.72 | 15.38 | 23.08 | 34.36 | 48.72 | 61.54 |

a sharp decrease in slump [36]. Additionally, a higher amount of glass powder and rice husk ash enhances the interaction forces between particles, hindering the flow of the paste and causing a rapid slump reduction. Although the slump decreased with increasing silicon-calcium ratio, the measured values ranged from 75 mm to 195 mm, which remain within the workable range for normal concrete applications.

Based on the data in Table 3, the relationship between the slump of WGRC and the silicon-calcium ratio was fitted, and the fitting result shows a power function decreasing trend. The fitting results are shown in Fig 6. The fitted results in Fig 6 can be referenced to understand the slump variation trend of WGRC.

### 3.2 Water absorption rate

The results of the water absorption rate experiment are shown in Fig 7. Overall, it can be observed that as the silicon-calcium ratio increases, the water absorption rate of WGRC gradually increases. This result contradicts previous reports [37–40]. Tang et al. [39] pointed out that an increase in the silicon-calcium ratio promotes a higher content of tricalcium silicate in the cement clinker. The hydration of tricalcium silicate generates more calcium hydroxide, which improves the density of the concrete and thus reduces the water absorption rate. In this study, the increase in silicon-calcium ratio also led to a higher dosage of rice husk ash, which has strong water absorption properties. Therefore, the water absorption rate showed an increasing trend. This behavior may be related to the porous structure and high specific surface area of rice husk ash, which can increase water absorption capacity despite potential densification effects. It should be noted that when the silicon-calcium ratio was low, the water absorption rate of WGRC did not show a significant increase, but at higher silicon-calcium ratios, the water absorption rate exhibited a noticeable rise.

For example, compared with the water absorption rate at a silicon-calcium ratio of 0.34, at a silicon-calcium ratio of 0.46, the water absorption rate increased by only 1.11%, 1.98%, and 1.71% after 7, 28, and 90 days of curing,

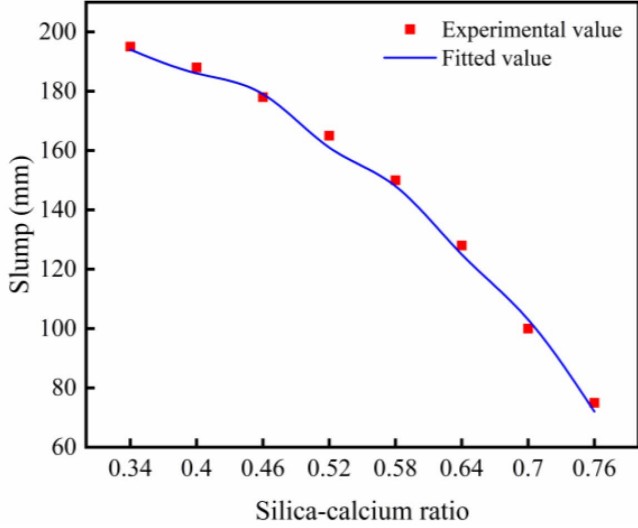

**Fig 6. Slump fitting results.**

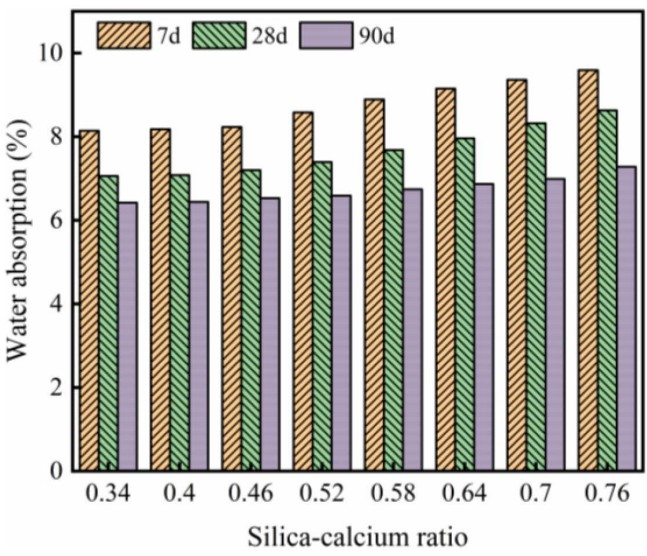

**Fig 7. Calculation results of water absorption.**

respectively. From Fig 7, it can also be seen that as the curing age increased, the water absorption rate gradually decreased. For instance, when the silicon-calcium ratio was 0.58, the water absorption rate at 7 days of curing was 8.89%, and at 90 days of curing, it decreased to 6.74%, a reduction of 24.18%. This result is consistent with previous reports [41,42]. Liu et al. [43] suggested that as the curing age increases, the capillary pores within the concrete are gradually filled with hydration products, which reduces the connectivity of the pores, making it more difficult for moisture to penetrate the concrete, thus lowering the water absorption rate.

The pearson correlation coefficient was calculated based on the water absorption rate data at different curing ages in Fig 7, and the results are shown in Table 4.

From Table 4, it can be observed that there is a significant positive correlation between the silicon-calcium ratio and the water absorption rate of WGRC at each curing age, with correlation coefficients close to 1. This indicates that an increase in the silicon-calcium ratio leads to an increase in the water absorption rate. Furthermore, regression analysis was conducted using the data from Fig 7, and the relationship between the water absorption rate, silicon-calcium ratio, and curing age was derived, as shown in Equation (1).

$$W = 8.696 - 0.014 \times C + 0.834 \times (Si/Ca) \quad (R^2 = 0.985) \tag{1}$$

Where: $W$ is the water absorption rate; $C$ is the curing age.

### 3.3 Compressive strength, splitting tensile strength, and flexural strength

The compressive strength test results of WGRC are shown in Fig 8. It is observed that at 7 days, the compressive strength decreases with the increase in silicon-calcium ratio. When the silicon-calcium ratio increases from 0.34 to 0.76, the compressive strength decreases from 23.02 MPa to 12.35 MPa. This indicates that at early ages, a lower silicon-calcium ratio has a higher hydration rate and better early strength development. This trend is similar to the reports by Wei et al. [40] and Hot et al. [44]. This is mainly because the higher Ca content promotes the rapid formation of C-S-H gel, enhancing the compactness of the early structure. Furthermore, with the increase in silicon-calcium ratio, the decrease in Ca content may reduce the formation rate of C-S-H gel, thereby affecting the early strength gain. A too high

**Table 4. Pearson correlation coefficient.**

| Curing age | 7d | 28d | 90d |
|---|---|---|---|
| Pearson correlation coefficient | 0.994 | 0.993 | 0.987 |

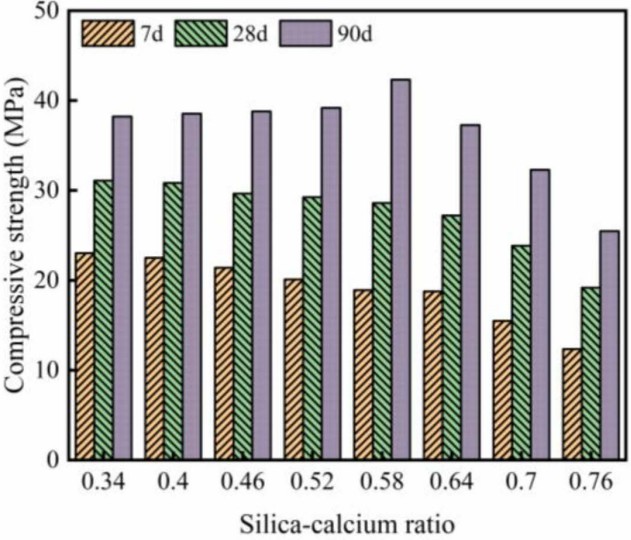

**Fig 8. Compressive strength.**

silicon-calcium ratio may result in insufficient framework support, leading to a decrease in structural stability and, consequently, a decrease in compressive strength.

At 28 days, the compressive strength still exhibits a similar trend, with low silicon-calcium ratio samples showing relatively higher compressive strength. For example, when the silicon-calcium ratio is 0.34, the compressive strength is 31.07 MPa. Additionally, compared to the 7-day age, the compressive strength of WGRC increases, and the compressive strength of samples with a silicon-calcium ratio between 0.46 and 0.58 approaches 30 MPa, indicating that this range may be more suitable for the silicon-calcium ratio. It is worth noting that for high silicon-calcium ratios (0.7 and above), the compressive strength is still relatively low, which may be due to the reduced stability of the gel structure and higher porosity [45].

At 90 days, the compressive strength of WGRC generally further increases, reaching the maximum value at a silicon-calcium ratio of 0.58. This indicates that a moderate silicon-calcium ratio can promote the stable growth of C-S-H gel, improving long-term strength. It should be noted that as the silicon-calcium ratio increases further, the compressive strength decreases. For example, when the silicon-calcium ratio is 0.76, the compressive strength is only 25.5 MPa. This may be due to the higher Si content leading to the gel structure being dominated by silicate phases, reducing its hardening performance, and the pore structure being more open, resulting in a decrease in overall compactness [46,47]. Furthermore, Hu et al. [48] also pointed out that insufficient Ca content affects the crosslinking degree of C-S-H gel, leading to a decrease in compressive strength.

The splitting tensile strength test results of WGRC are shown in Fig 9. As seen in Fig 9, at 7 days, the splitting tensile strength of WGRC decreases gradually with the increase in silicon-calcium ratio, which is similar to the trend observed for compressive strength. This may be due to the fact that at early ages, samples with a lower silicon-calcium ratio have a higher calcium content, which promotes the rapid formation of C-S-H gel and enhances the bonding strength of the interfacial transition zone, thus improving tensile strength [49]. On the other hand, samples with a higher silicon-calcium ratio have insufficient calcium content, leading to slower growth of C-S-H gel, which results in lower splitting tensile strength.

At 28 days, the trend in splitting tensile strength differs from that at 7 days, with a peak value appearing at a silicon-calcium ratio of 0.52. This suggests that, at this stage, the growth and structure of C-S-H gel have been optimized, leading

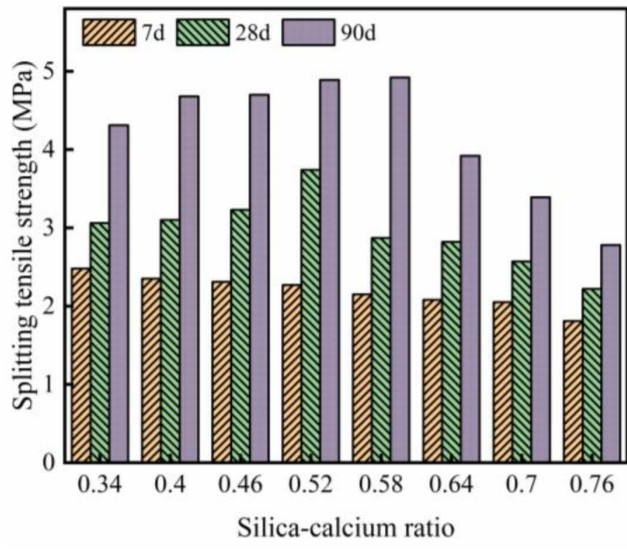

**Fig 9. Splitting tensile strength.**

to better splitting tensile strength. Additionally, for low silicon-calcium ratio samples, the excessive calcium content may lead to insufficient crosslinking of C-S-H gel, resulting in a decrease in the compactness of the microstructure [50]. When the silicon-calcium ratio exceeds 0.58, the splitting tensile strength decreases significantly, indicating that a higher silicon-calcium ratio may weaken the interfacial transition zone, thereby reducing tensile strength.

At 90 days, the peak value of splitting tensile strength occurs at a silicon-calcium ratio of 0.58, slightly delayed compared to 28 days. This suggests that as the curing age increases, the splitting tensile strength of samples with a high silicon-calcium ratio gradually surpasses that of low silicon-calcium ratio samples due to the continuous development of C-S-H gel. It should be noted that when the silicon-calcium ratio exceeds 0.58, the splitting tensile strength gradually decreases. This phenomenon is similar to the results reported by Chen et al. [51]. Chen et al. [51] suggested that this might be due to insufficient calcium content in high silicon-calcium ratio samples during long-term curing, which reduces the crosslinking capacity of C-S-H gel, making it difficult to effectively repair microcracks, thus affecting splitting tensile strength. It is important to emphasize that this study further extended the diversity of silicon elements by incorporating glass powder and rice husk ash, causing the optimal range of silicon-calcium ratio to shift to the right.

The bending strength test results of WGRC are shown in Fig 10. As observed, at 7 days, the bending strength gradually decreases with the increase in silicon-calcium ratio. This trend suggests that at early curing stages, samples with lower silicon-calcium ratios have a higher concentration of calcium ions, which promotes faster formation of early C-S-H gel and a denser interfacial transition zone, thus enhancing the bending strength of the concrete. However, with the increase in silicon-calcium ratio, the decrease in calcium content leads to a slower early formation rate of C-S-H gel, resulting in a more porous interfacial transition zone, which affects the bending strength.

At 28 days, the trend of bending strength changes, with the peak value occurring at a silicon-calcium ratio of 0.46. Compared to 7 days, the bending strength of WGRC increases, but the growth rate varies with different silicon-calcium ratios. The bending strength of lower silicon-calcium ratio samples (0.34 and 0.4) remains relatively high (4.47 MPa and 4.68 MPa, respectively), but the increase is smaller compared to samples with a silicon-calcium ratio of 0.46. This is because, at lower silicon-calcium ratios, C-S-H gel has already formed to a sufficient extent during the early stage, resulting in a more gradual growth in the later stage, whereas at a silicon-calcium ratio of 0.46, the C-S-H gel continues to develop at 28 days, improving the interfacial transition zone and leading to the maximum bending strength.

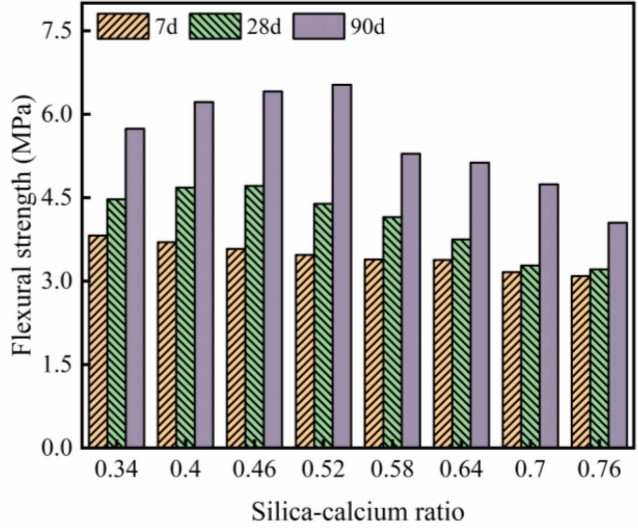

**Fig 10. Flexural strength.**

At 90 days, the peak bending strength is further delayed, occurring at a silicon-calcium ratio of 0.52. As the curing age increases, the bending strength of samples with silicon-calcium ratios of 0.46 and 0.52 improves due to the continued growth of C-S-H gel and the further enhancement of the interfacial transition zone. However, when the silicon-calcium ratio exceeds 0.58, the bending strength decreases significantly (e.g., 4.05 MPa at a silicon-calcium ratio of 0.76). This may be because, during long-term curing, samples with higher silicon-calcium ratios have lower calcium content, leading to insufficient growth of C-S-H gel, resulting in a more porous microstructure that affects the bending capacity of the concrete.

The reduction in early-age strength at higher Si/Ca ratios is mainly related to the slower reaction of glass powder and rice husk ash compared with cement hydration. At early ages, the increase in Si/Ca ratio reduces the relative calcium availability in the system, while the silica-rich supplementary materials have not yet reacted sufficiently to form enough binding products. As a result, the formation of strength-contributing hydration products is limited, leading to lower early strength. With increasing curing age, the pozzolanic activity of glass powder and rice husk ash becomes more evident. The active $SiO_2$ gradually reacts with $Ca(OH)_2$ to form additional C-S-H gel, which improves matrix compactness and enhances later-age strength. However, when the Si/Ca ratio becomes excessively high, the available calcium in the system is insufficient to sustain the formation of a more effective continuous gel network. Therefore, later-age strength is improved at moderate Si/Ca ratios rather than increasing continuously with increasing Si/Ca ratio.

### 3.4 Gray correlation analysis

Gray relational analysis was adopted due to its suitability for multi-indicator evaluation with limited sample size and without strict requirements on data distribution. First, the correlation coefficients are calculated, as shown in Equation (2). Then, the weighted average of the correlation coefficients is taken to obtain the correlation degree, and the results are presented in Table 5.

$$\xi_i\left(k\right) = \frac{\min_i\min_k\left|x_0\left(k\right) - x_i\left(k\right)\right| + \rho\max_i\max_k\left|x_0\left(k\right) - x_i\left(k\right)\right|}{\left|x_0\left(k\right) - x_i\left(k\right)\right| + \rho\max_i\max_k\left|x_0\left(k\right) - x_i\left(k\right)\right|}$$

(2)

Where: $x_0(k)$ is the reference sequence; $x_i(k)$ is the comparison sequence; $\rho$ is the distinguishing coefficient, set to 0.5.

As shown in Table 5, the grey relational analysis effectively reflects the impact of the silicon-to-calcium ratio on the mechanical properties of WGRC. The comprehensive grey relational degrees are highest when the silicon-to-calcium ratio is 0.34 and 0.4, indicating that the overall mechanical performance is optimal for these mix ratios. In addition, the comprehensive grey relational degrees for 0.46 and 0.52 are also relatively high, which suggests these are also within the optimal range. However, when the silicon-to-calcium ratio is 0.58 or higher, the overall mechanical performance starts to decline,

**Table 5. Calculation results of grey correlation degree.**

| Silica-calcium ratio | Compressive strength | Splitting tensile strength | Flexural strength | Comprehensive correlation degree |
|---|---|---|---|---|
| 0.34 | 0.8908 | 0.7216 | 0.7895 | 0.8006 |
| 0.40 | 0.8539 | 0.6934 | 0.8380 | 0.7951 |
| 0.46 | 0.7592 | 0.6971 | 0.8384 | 0.7649 |
| 0.52 | 0.7132 | 0.8625 | 0.7371 | 0.7709 |
| 0.58 | 0.7569 | 0.6567 | 0.5105 | 0.6414 |
| 0.64 | 0.5955 | 0.4750 | 0.4539 | 0.5081 |
| 0.70 | 0.4402 | 0.4144 | 0.3698 | 0.4081 |
| 0.76 | 0.3333 | 0.3333 | 0.3333 | 0.3333 |

particularly for ratios of 0.7 and 0.76, which have the lowest relational degrees, indicating poor mechanical performance. Overall, the silicon-to-calcium ratio range of 0.34~0.52 is considered ideal, as it exhibits good compressive, tensile, and flexural strength in concrete, making it suitable for recommending as a mix ratio.

### 3.5 Calcium-Dilicon ratio in hydrated calcium silicate

Using EDS experiments, the internal hydrated calcium silicate (C-S-H gel) of WGRC was analyzed at curing ages of 7 days, 28 days, and 90 days. The atomic percentages of elements such as Ca, Si, and O were obtained, and by calculating the ratio of the atomic percentage of Si to that of Ca, the calcium-silicon ratio in the internal C-S-H gel of WGRC was determined. The results are shown in Table 6.

From Table 6, it can be observed that the regulation of the calcium-silicon ratio significantly affects the calcium-silicon ratio in the internal C-S-H gel of WGRC at different curing ages. Based on the atomic percentages of Si and Ca, the Si/Ca ratio of the gel phase was calculated, and the results are summarized in Table 6. It can be observed that the gel-phase Si/Ca ratio increased consistently with both the designed mixture Si/Ca ratio and curing age. For example, at 7 days, the gel-phase Si/Ca ratio increased from 1.22 for the 0.34 mixture to 1.72 for the 0.76 mixture, while at 90 days it further increased to 1.36 and 1.82, respectively. This indicates that the incorporation of glass powder and rice husk ash continuously promoted the enrichment of silicate species in the hydration products during curing.

However, the increase in gel-phase Si/Ca ratio did not lead to a monotonic improvement in macroscopic performance. Instead, the compressive strength, splitting tensile strength, flexural strength, and shear strength generally exhibited optimal values within a moderate mixture Si/Ca range, rather than at the highest Si/Ca level. This suggests that the role of Si/Ca regulation is not merely to increase the silicon content of the gel, but to optimize the balance between gel polymerization, reaction continuity, and calcium supply [52].

## 4. Shear behavior

### 4.1 Normal stress and shear stress

The normal and shear stresses were calculated using Equation (3) [53], and the results are shown in Fig 11.

$$\sigma = \frac{P \cdot \cos\theta}{A}, \tau = \frac{P \cdot \sin\theta}{A}$$

(3)

Where: $\sigma$ is the normal stress; $\tau$ is the shear stress; $P$ is the maximum load sustained by the specimen; $\theta$ is the shear angle; $A$ is the cross-sectional area.

**Table 6. Silica-calcium ratio of C-S-H gel in WGRC.**

| Silica-calcium ratio | Curing age | | |
|---|---|---|---|
| | 7d | 28d | 90d |
| 0.34 | 1.22 | 1.30 | 1.36 |
| 0.40 | 1.27 | 1.33 | 1.39 |
| 0.46 | 1.35 | 1.38 | 1.45 |
| 0.52 | 1.41 | 1.47 | 1.52 |
| 0.58 | 1.50 | 1.55 | 1.60 |
| 0.64 | 1.58 | 1.62 | 1.67 |
| 0.70 | 1.63 | 1.71 | 1.75 |
| 0.76 | 1.72 | 1.78 | 1.82 |

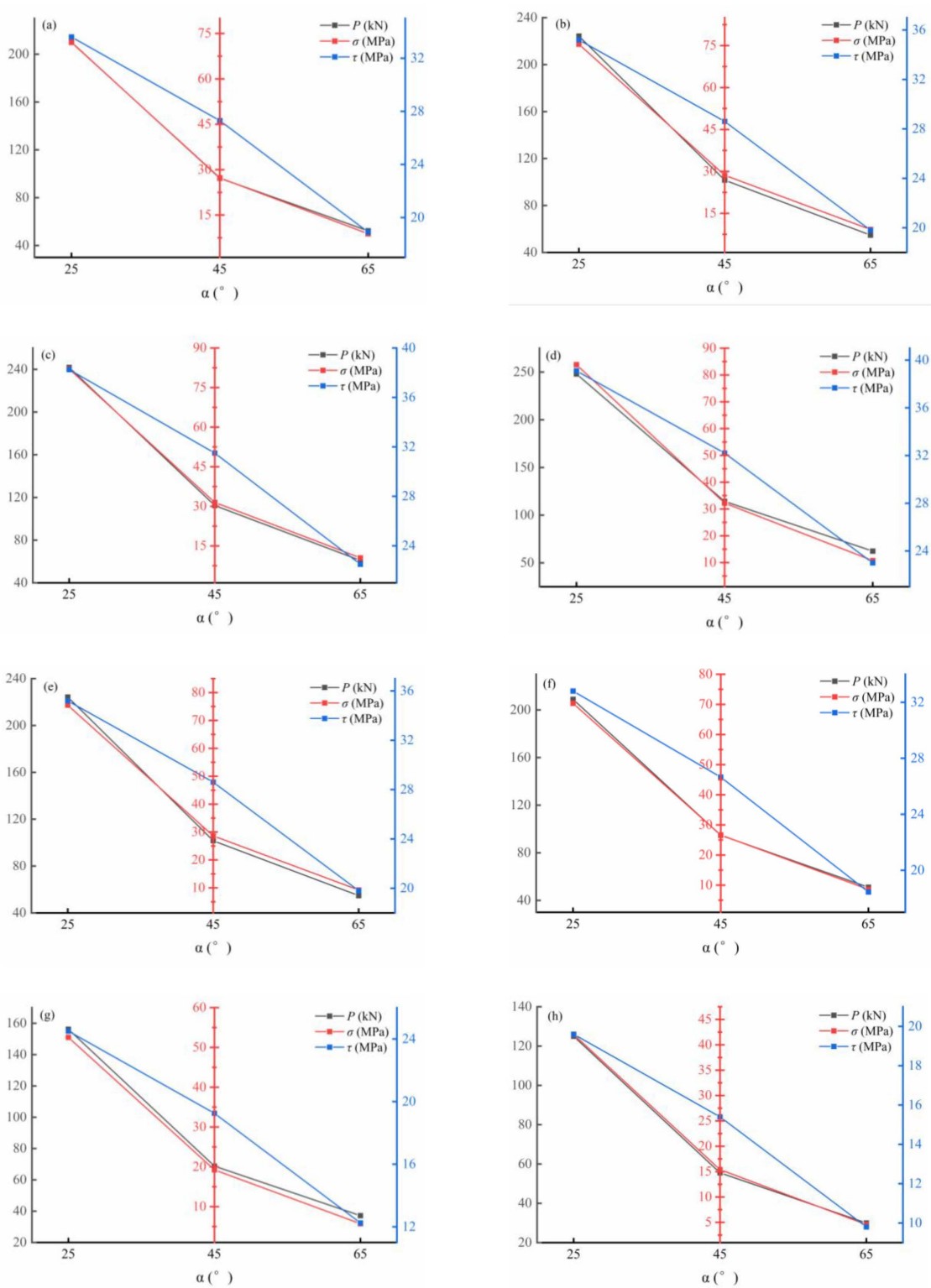

**Fig 11. Normal stress and shear stress of samples with different silicon-calcium ratios: (a) 0.34, (b) 0.4, (c) 0.46, (d) 0.52, (e) 0.58, (f) 0.64, (g) 0.7, (h) 0.76.**

*In general, as the shear angle decreased, both the normal stress and shear stress increased significantly, indicating strong sensitivity to the loading angle. Notably, at a shear angle of 25°, the loading direction approached uniaxial compression, resulting in stress concentration on the shear plane. The enhanced normal compression improved interfacial friction and bond strength, leading to the highest observed shear stress. Moreover, the Si/Ca ratio had a pronounced influence on the shear response. For example, at a shear angle of 25°, the shear stress initially increased and then decreased with rising Si/Ca ratio, reaching a peak value of 39.10 MPa at a ratio of 0.52. A similar trend was observed for the normal stress. These results indicate that WGRC exhibited optimal shear-bearing capacity within a moderate Si/Ca ratio range of 0.46–0.52. The same behavior was consistently observed at shear angles of 45° and 65°, further confirming that the shear strength was predominantly governed by the Si/Ca ratio.*

## 4.2 Cohesion and internal friction angle

Based on the Mohr-Coulomb criterion [54], the cohesion (*c*) and internal friction angle (*φ*) were obtained through regression fitting, as defined in Equation (4), with results summarized in Table 7.

$$\tau = c + \sigma \cdot \tan\varphi$$

(4)

From Table 7, it is evident that both cohesion and internal friction angle exhibited significant variations with the Si/Ca ratio, although their response patterns differed. Cohesion showed a distinct peak-shaped nonlinear trend with increasing Si/Ca ratio, while the internal friction angle remained relatively stable, showing only a slight increase under high Si/Ca conditions.

Specifically, the cohesion values at low Si/Ca ratios of 0.34 and 0.40 were 23.46 MPa and 24.57 MPa, respectively, representing moderate levels. As the Si/Ca ratio increased, the degree of cementitious reaction was enhanced, leading to a substantial increase in cohesion. At Si/Ca ratios of 0.46 and 0.52, the values reached 27.29 MPa and 27.90 MPa, respectively, indicating the formation of a denser and more continuous gel structure, which contributed to improved interfacial bonding and shear resistance. However, when the Si/Ca ratio further increased to 0.70 and 0.76, cohesion dropped sharply to 16.21 MPa and 12.97 MPa, respectively-a decline of over 40%. This suggests that an excessively high Si/Ca ratio results in calcium deficiency, impeding complete gelation. The accumulation of unreacted glass powder and rice husk ash particles disrupted the continuity of the interfacial transition zone, thereby weakening the material's cohesive strength.

It is worth noting that the internal friction angle showed limited fluctuation, mostly ranging between 7.61° and 8.01°, implying that shear failure was primarily governed by bond degradation rather than particle interlocking. A slight increase to 8.96° was observed at Si/Ca ratios of 0.70 and 0.76, possibly due to increased roughness of the fracture surfaces.

Importantly, the internal friction angle remained below 9°, which is significantly lower than the typical values of 30°-45° reported for conventional concrete dominated by natural coarse aggregates [55,56]. This indicates that the failure mechanism of such green cementitious composites is primarily governed by matrix–interface bond strength, rather than interparticle friction-a notable distinction from traditional concrete systems.

**Table 7. Cohesion and internal friction angle.**

| Si/Ca | 0.34 | 0.40 | 0.46 | 0.52 | 0.58 | 0.64 | 0.70 | 0.76 |
|---|---|---|---|---|---|---|---|---|
| *c*/MPa | 23.46 | 24.57 | 27.29 | 27.9 | 24.57 | 22.9 | 16.21 | 12.97 |
| *φ*/° | 8.01 | 8.01 | 7.61 | 7.61 | 8.01 | 8.01 | 8.96 | 8.96 |

## 4.3 Shear stress-strain relationship

The shear stress-strain curves of WGRC are presented in Fig 12. Overall, all curves exhibited pronounced nonlinear shear responses, typically consisting of a rapid linear ascending stage, a peak plateau, and a post-peak softening phase. The combined effect of shear angle and Si/Ca ratio had a significant influence on peak shear stress, strain ductility, and shear brittleness.

Regarding the influence of shear angle, under the same Si/Ca conditions, the 25° specimens consistently exhibited the highest peak shear stress, followed by those with 45° and then 65° angles. From the perspective of Si/Ca ratio, the peak shear stress of WGRC increased steadily as the Si/Ca ratio rose from 0.34 to 0.58, demonstrating a clear strength enhancement trend. However, further increases to 0.70 and 0.76 led to a decline in strength, indicating the existence of an optimal Si/Ca range, namely 0.52–0.58. This trend closely matched that observed for compressive strength, highlighting the regulatory effect of Si/Ca ratio on hydration progression and gel compactness.

It should be noted that in terms of shear ductility, high Si/Ca ratio specimens (0.70 and 0.76) exhibited slower softening behavior and longer post-peak deformation plateaus despite their lower peak shear stresses, reflecting a more ductile failure mode. In contrast, moderate Si/Ca specimens (e.g., 0.52 and 0.58) showed the highest strengths but also exhibited steeper post-peak drops and more pronounced shear brittleness. These findings suggest that while increasing the Si/Ca ratio beyond the optimal range weakens the structural strength, the formation of low-polymerization silicate gels may enhance deformability and crack-bridging capacity.

## 4.4 Shear damage constitutive model

### 4.4.1 Model formulation.

(1) Theoretical basis

As illustrated in Fig 12, the shear stress-strain response of WGRC exhibits a clearly nonlinear behavior under shear loading. The pre-peak stage is approximately linear, while the post-peak phase displays a softening trend due to progressive microcrack development. To characterize the damage evolution process, a shear damage constitutive model was developed based on the framework of Continuum Damage Mechanics (CDM) [57]. In this model, material degradation under shear is described using a stress-reduction approach, incorporating material composition and loading path as critical control parameters.

CDM theory introduces a scalar damage variable $D$, defined as:

$$D = 1 - \frac{A}{A_0} \tag{5}$$

Where: $A_0$ is the initial effective load-bearing area, and $A$ is the current effective load-bearing area.

Based on the effective stress principle, the shear stress-strain relationship can be expressed as:

$$\tau = (1 - D) \cdot \tau_0 \tag{6}$$

Where: $\tau$ is the current shear stress, and $\tau_0$ is the peak shear stress under the undamaged state.

(2) Damage variable

Previous studies by Sima et al. [58] and Wu et al. [59] have suggested that the damage evolution in concrete under monotonic loading can be described by a nonlinear function driven by shear strain. In this study, an exponential form is adopted:

$$D = 1 - \exp\left[-\left(\frac{\gamma}{\gamma_0}\right)^n\right] \tag{7}$$

**Fig 12. Shear stress-shear strain curves of samples with different silicon-calcium ratios: (a) 0.34, (b) 0.4, (c) 0.46, (d) 0.52, (e) 0.58, (f) 0.64, (g) 0.7, (h) 0.76.**

Where: $\gamma$ is the current shear strain; $\gamma_0$ is the critical shear strain corresponding to the peak shear stress; $n$ is the damage evolution index, which controls the steepness of the transition from the elastic stage to the softening stage.

By substituting the above into the stress formulation, the shear stress can be expressed as:

$$\tau = \left[1 - \exp\left(-\left(\frac{\gamma}{\gamma_0}\right)^n\right)\right] \cdot \tau_0 \tag{8}$$

(3) Multi-parameter coupling

As shown in the above expression, the two key parameters $\tau_0$ and $n$ play a decisive role in determining the material response. To enhance the model's adaptability to various material states and loading paths, both parameters were constructed as functions of Si/Ca ratio, shear angle, and 28-day compressive strength. This approach allows for the coupling of WGRC composition and macro-mechanical properties in the damage description.

According to the Mohr-Coulomb theory, the peak shear stress is determined by both cohesion and normal stress. In this study, to avoid the explicit use of normal stress under different shear angles, an empirical strength function was proposed:

$$\tau_0 = \alpha \cdot f_c \cdot k \tag{9}$$

Where: $\alpha$ is a shear-compression conversion coefficient (taken as 0.15); $f_c$ is the 28-day compressive strength; $k$ is a correction function defined as:

$$k = \alpha_0 + \alpha_1 \cdot \cos(\theta) + \alpha_2 \cdot \beta + \alpha_3 \cdot \beta^2 \tag{10}$$

Where: $\beta$ is the Si/Ca ratio.

The damage index $n$ reflects the rate of softening and the degree of shear brittleness, which is governed by structural compactness and loading path. The following empirical relationship was proposed:

$$n = b_0 + b_1 \cdot \theta + b_2 \cdot \beta + b_3 \cdot f_c \tag{11}$$

(4) Final formulation

Based on the above derivation, the complete expression of the proposed shear damage model is:

$$\tau = \left[1 - \exp\left(-\left(\frac{\gamma}{\gamma_0}\right)^{(b_0 + b_1 \cdot \theta + b_2 \cdot \beta + b_3 \cdot f_c)}\right)\right] \cdot \alpha \cdot f_c \cdot \left(\alpha_0 + \alpha_1 \cdot \cos(\theta) + \alpha_2 \cdot \beta + \alpha_3 \cdot \beta^2\right) \tag{12}$$

This model simultaneously incorporates the effects of microstructural composition and loading configuration, offering strong expandability, physical relevance, and engineering applicability.

**4.4.2 Parameter determination.** Parameters such as $f_c$, $\gamma$, and $\gamma_0$ were directly obtained from experimental data and are not repeated here. The coefficient $\alpha$ was taken as 0.15. The parameters $b_0$, $b_1$, $b_2$, $b_3$ and $\alpha_0$, $\alpha_1$, $\alpha_2$, $\alpha_3$ were determined through regression fitting. Taking the 45° shear angle as an example, the fitted results are summarized in Table 8.

**4.4.3 Model validation.** The theoretical prediction curves were compared with representative experimental data, as shown in Fig 13. The results indicate that the model provides an excellent fit across the entire loading process, particularly in capturing the peak shear stress, which closely matches the experimental observations. This confirms the model's ability to effectively predict the ultimate strength behavior of concrete under varying Si/Ca ratios and shear angles.

**Table 8. Parameter fitting results.**

| Sample | Parameters | | | | | | | |
|--------|------|------|------|------|------|------|------|------|
| | $b_0$ | $b_1$ | $b_2$ | $b_3$ | $\alpha_0$ | $\alpha_1$ | $\alpha_2$ | $\alpha_3$ |
| S/C-0.34 | 1.012 | −0.0154 | 0.513 | −0.021 | 0.654 | 0.253 | −0.405 | 0.302 |
| S/C-0.40 | 1.021 | −0.0174 | 0.485 | −0.019 | 0.696 | 0.274 | −0.386 | 0.316 |
| S/C-0.46 | 1.033 | −0.0165 | 0.526 | −0.024 | 0.715 | 0.302 | −0.294 | 0.298 |
| S/C-0.52 | 0.988 | −0.0197 | 0.714 | −0.032 | 0.654 | 0.288 | −0.314 | 0.279 |
| S/C-0.58 | 1.213 | −0.0226 | 0.863 | −0.016 | 0.814 | 0.311 | −0.367 | 0.255 |
| S/C-0.64 | 1.402 | −0.0433 | 0.652 | −0.029 | 0.866 | 0.365 | −0.411 | 0.306 |
| S/C-0.70 | 1.321 | −0.0614 | 0.511 | −0.028 | 0.723 | 0.316 | −0.399 | 0.311 |
| S/C-0.76 | 1.524 | −0.0421 | 0.499 | −0.022 | 0.598 | 0.563 | −0.403 | 0.257 |

However, some deviations were observed in the prediction of shear strain. Specifically, the model slightly underestimates the peak strain, which may be attributed to the limitations in the damage evolution function in describing the post-peak softening process. Furthermore, the theoretical curves are inherently smooth and continuous, whereas the experimental curves exhibit noticeable fluctuations. These fluctuations likely result from local perturbations caused by interfacial slip and unstable microcrack propagation, which are not captured in the current analytical model.

Despite these limitations, the model consistently demonstrates high predictive accuracy across a wide range of Si/Ca ratios, suggesting that the inclusion of shear angle, Si/Ca ratio, and compressive strength as key variables successfully captures the coupling effects between the macro- and micro-structural characteristics of WGRC. Overall, the proposed shear damage constitutive model exhibits strong adaptability and physical reliability.

## 5. Microstructural analysis

Fig 14 presents the microstructural morphology of WGRC samples with different Si/Ca ratios at a curing age of 28 days. Overall, the primary hydration products observed in all samples are amorphous calcium silicate hydrate (C-S-H) gels and needle-like ettringite (AFt) crystals. In addition, visible cracks, pores, and some residual unhydrated particles are also present.

Under low Si/Ca conditions, a large quantity of flocculent and block-like C-S-H gels was formed. These randomly distributed but relatively dense gels were extensively deposited between cement particles and around aggregates, effectively enhancing the structural integrity and bonding performance of the matrix. At this stage, AFt crystals were still underdeveloped, with only a few fine, short needle-like structures observed in localized regions. This indicates that the availability of aluminum and sulfate ions in the reaction environment was insufficient to promote significant AFt precipitation.

In samples with moderate Si/Ca ratios (e.g., 0.52, 0.58, and 0.64), a typical "gel-crystal synergy" structure was observed. The content of AFt crystals increased markedly while C-S-H gels remained well-distributed and dense. The AFt crystals were primarily deposited in a clustered or radial form within capillary pores and the interfacial transition zones, while the C-S-H gels continued to dominate the matrix cohesion. The hydration products at this stage exhibited an orderly and complementary spatial arrangement, forming a robust microstructural load-bearing system, which is consistent with the observed improvements in macroscopic strength.

At high Si/Ca ratios, both the size and amount of AFt crystals further increased, leading to dense and widespread deposition. However, the quantity of C-S-H gels was significantly reduced, and their distribution became discontinuous, resulting in a notable increase in porosity. It is worth noting that obvious cracks and voids were found in multiple regions, indicating that the limited availability of Ca²⁺ under high Si/Ca conditions led to incomplete hydration, thereby weakening the structural stability of the cementitious matrix. Furthermore, the excessive proportion of AFt crystals may contribute to brittleness, facilitating the initiation and propagation of local microcracks, which ultimately impairs the material's ductility.

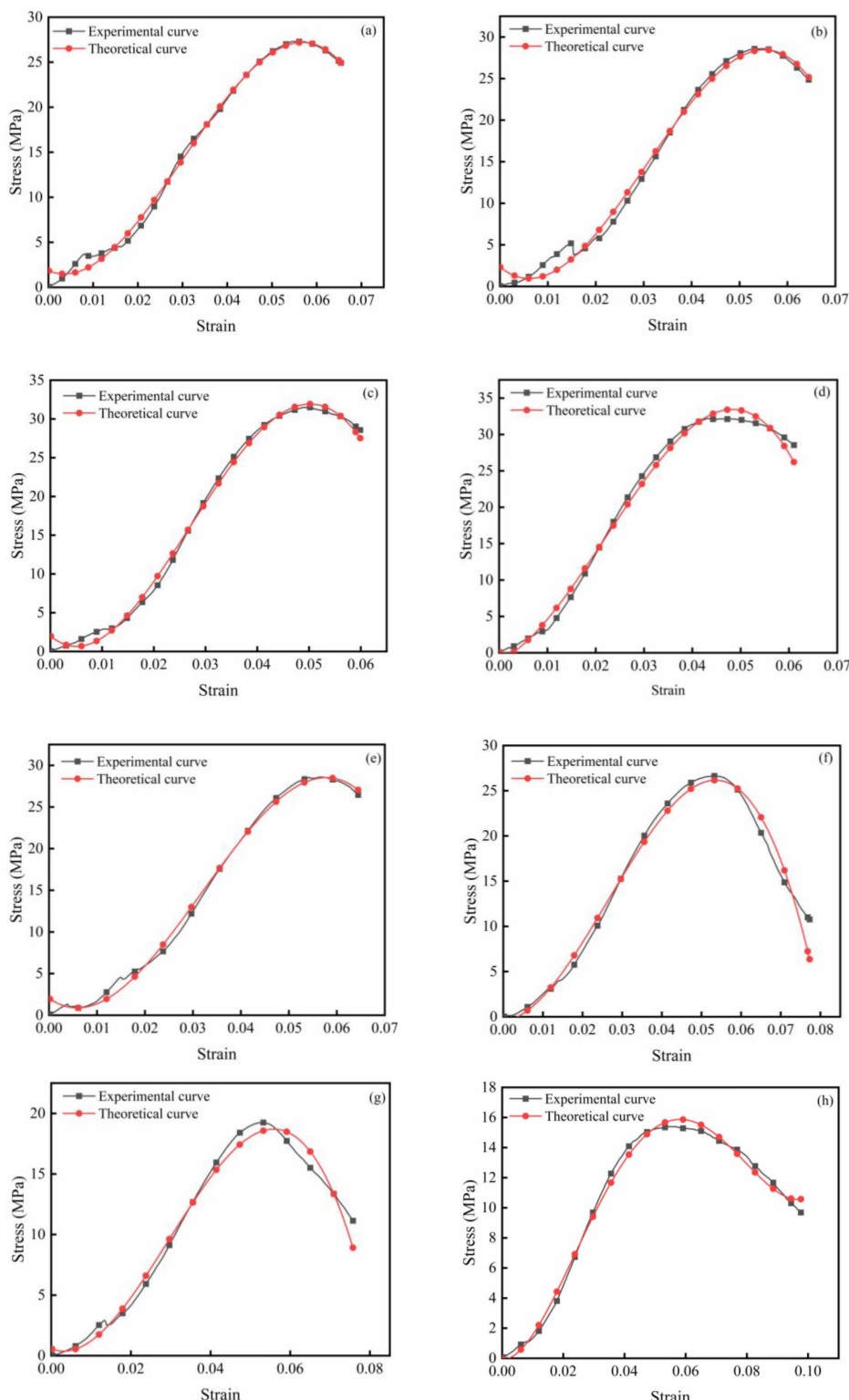

**Fig 13. Comparison of experimental curves and theoretical curves of samples with different silicon-calcium ratios: (a) 0.34, (b) 0.4, (c) 0.46, (d) 0.52, (e) 0.58, (f) 0.64, (g) 0.7, (h) 0.76.**

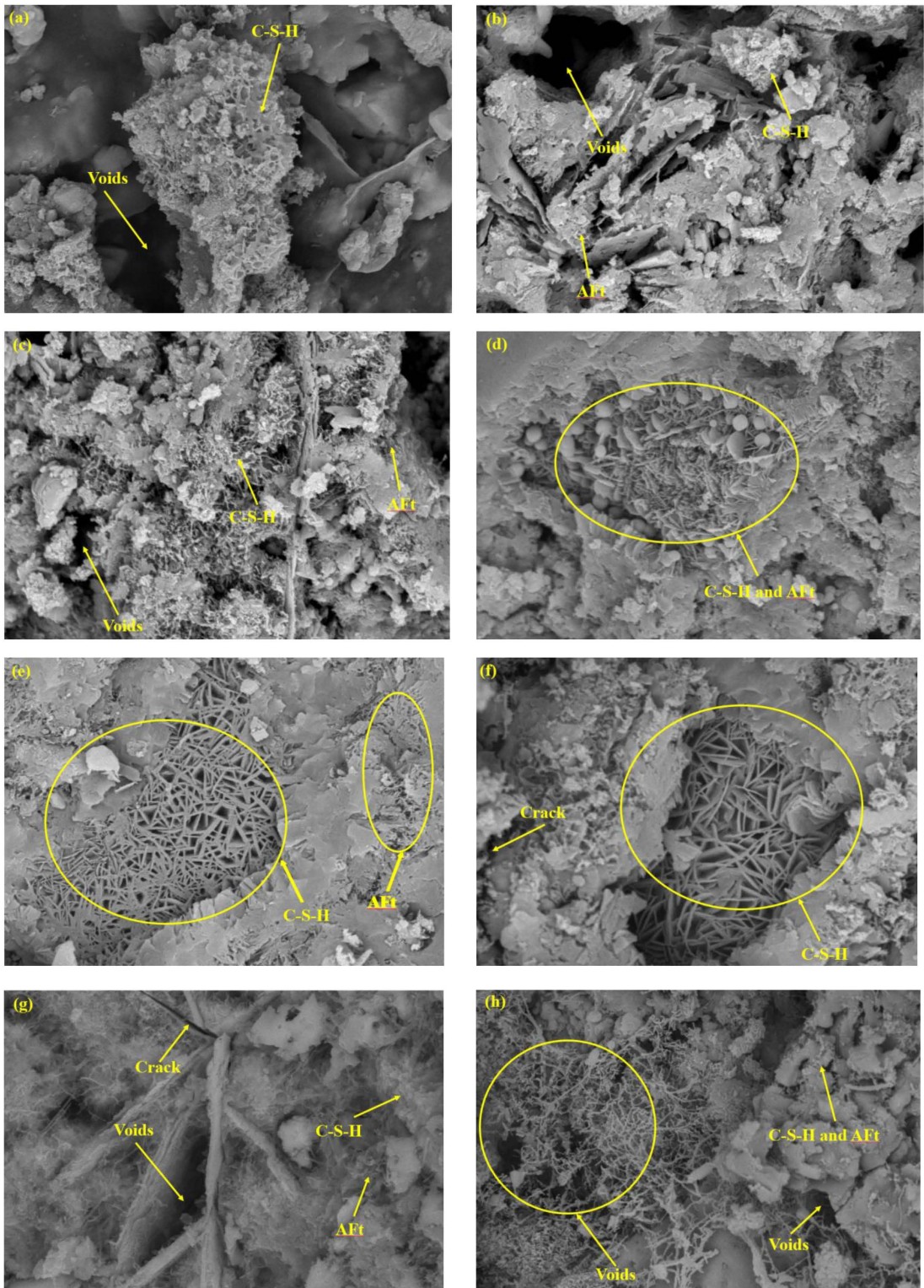

**Fig 14. SEM morphologies of samples with different silicon-calcium ratios: (a) 0.34, (b) 0.4, (c) 0.46, (d) 0.52, (e) 0.58, (f) 0.64, (g) 0.7, (h) 0.76.**

# 6.  Durability performance

## 6.1  Rapid chloride ion penetration

The rapid chloride ion penetration test (RCPT) was conducted to measure the electric charge passed through WGRC within 6 hours to assess its resistance to chloride ion penetration. The chloride ion penetration level was classified according to the standard in ASTM C 1202. The results of the electric charge test are shown in Table 9.

The RCPT results indicate that as the curing age increases, the electric charge passed through WGRC gradually decreases, which suggests a reduction in chloride ion permeability and an increase in WGRC's density. This phenomenon is mainly attributed to the ongoing hydration reaction, which causes the C-S-H gel to continuously fill the pores, enhancing the resistance to penetration. Moreover, the effect of the silica-calcium ratio on the RCPT results exhibits a clear "U" shaped trend. Specifically, within the moderate silica-calcium ratio range (0.52–0.58), WGRC shows the lowest electric charge passed, while both very low and high silica-calcium ratios lead to higher electric charge values. It is worth noting that although low silica-calcium ratio samples exhibit higher early strength (due to a faster hydration reaction), the C-S-H gel structure is relatively loose, and the pores have not been fully filled, resulting in higher chloride ion permeability and higher electric charge. As the silica-calcium ratio increases, the formation of the C-S-H gel becomes more stable, pore volume decreases, and the electric charge decreases gradually. However, when the silica-calcium ratio continues to increase beyond 0.64, the calcium content in the C-S-H gel is relatively low, which may reduce the stability of the gel products, causing a decline in resistance to penetration. Additionally, the structure of the C-S-H gel tends to be more silicate-based, which may reduce the density and increase microcracking, thereby increasing chloride ion permeability and resulting in a higher RCPT value.

## 6.2  Carbonation

Table 10 presents the results of the carbonation depth experiment. It was observed that the carbonation depth gradually increased with the carbonation time, but samples with different silica-calcium ratios showed significant differences. The carbonation depth of low silica-calcium ratio samples was consistently larger, while high silica-calcium ratio samples exhibited relatively smaller carbonation depths. This is mainly because the C-S-H gel structure in low silica-calcium ratio samples is looser, with a higher porosity, allowing $CO_2$ to penetrate faster, which leads to the rapid consumption of calcium in the cement and a larger carbonation depth. Additionally, high silica-calcium ratio samples, due to the relatively lower calcium content, initially lack the buffering capacity against $CO_2$, resulting in a faster increase in carbonation depth [60–62]. This result is consistent with the trends observed in the rapid chloride ion penetration experiment, indicating that optimizing the silica-calcium ratio helps improve the durability of WGRC.

Table 9.  Electric flux.

| Silica-calcium ratio | Curing age | | |
|---|---|---|---|
| | 7d | 28d | 90d |
| 0.34 | 4500 | 3800 | 3200 |
| 0.40 | 4200 | 3500 | 2900 |
| 0.46 | 3900 | 3100 | 2500 |
| 0.52 | 3500 | 2700 | 2100 |
| 0.58 | 3300 | 2500 | 1900 |
| 0.64 | 3700 | 2900 | 2300 |
| 0.70 | 4100 | 3400 | 2800 |
| 0.76 | 4600 | 4000 | 3400 |

**Table 10. Carbonization depth of WGRC at different carbonization times (Unit: mm).**

| Silica-calcium ratio | Carbonization time | | | |
|---|---|---|---|---|
| | 3d | 7d | 14d | 28d |
| 0.34 | 2.3 | 4.1 | 6.5 | 9.2 |
| 0.40 | 2.0 | 3.6 | 5.7 | 8.1 |
| 0.46 | 1.8 | 3.2 | 5.1 | 7.3 |
| 0.52 | 1.5 | 2.7 | 4.3 | 6.1 |
| 0.58 | 1.3 | 2.3 | 3.6 | 5.2 |
| 0.64 | 1.4 | 2.5 | 3.9 | 5.5 |
| 0.70 | 1.7 | 3.0 | 4.7 | 6.8 |
| 0.76 | 2.0 | 3.4 | 5.3 | 7.6 |

The carbonation coefficient is calculated based on the carbonation depth experiment results using the formula shown in Equation (13) [63,64].

$$k = \frac{d}{\sqrt{t}}$$

(13)

where $k$ is the carbonation coefficient, $d$ is the carbonation depth, $t$ is the carbonation time in days.

The carbonation coefficient calculation results are presented in Table 11. It can be observed that the carbonation rate of WGRC shows a trend of fast initial carbonation followed by a slowdown. The carbonation rate changes quickly in the early stages (3 days and 14 days), and then gradually stabilizes from 14 days to 28 days. Similar results have been reported in previous studies on carbonation coefficients of different types of concrete [65–67]. Gruyaert et al. [66] suggested that this is primarily due to the high content of $Ca(OH)_2$ in the early stage of carbonation, allowing $CO_2$ to diffuse quickly and react with $Ca(OH)_2$, leading to a rapid increase in carbonation depth. However, as the carbonation layer forms, the diffusion channels for $CO_2$ are restricted, which slows down the carbonation rate. From the perspective of samples with different silica-calcium ratios, low silica-calcium ratio samples consistently exhibit a higher carbonation rate, while high silica-calcium ratio samples show a faster carbonation rate initially but experience a more significant decrease in the later stages. It is noteworthy that the carbonation rate is the lowest when the silica-calcium ratio is 0.58, indicating that its C-S-H gel structure is the most compact. This structure effectively hinders $CO_2$ penetration, reduces the consumption rate of calcium, and thus improves the resistance to carbonation.

**Table 11. Carbonization coefficients of WGRC at different carbonization times (Unit: mm/√d).**

| Silica-calcium ratio | Carbonization time | | | | Mean value |
|---|---|---|---|---|---|
| | 3d | 7d | 14d | 28d | |
| 0.34 | 1.33 | 1.55 | 1.74 | 1.74 | 1.59 |
| 0.40 | 1.16 | 1.36 | 1.52 | 1.53 | 1.39 |
| 0.46 | 1.04 | 1.21 | 1.36 | 1.38 | 1.25 |
| 0.52 | 0.87 | 1.02 | 1.15 | 1.15 | 1.05 |
| 0.58 | 0.75 | 0.87 | 0.96 | 0.98 | 0.89 |
| 0.64 | 0.81 | 0.94 | 1.04 | 1.04 | 0.96 |
| 0.70 | 0.98 | 1.14 | 1.25 | 1.29 | 1.16 |
| 0.76 | 1.15 | 1.29 | 1.42 | 1.44 | 1.32 |

The pH value of the samples after carbonation was measured, and the results are shown in Table 12. From the table, it can be seen that during the early stage of carbonation (3 days and 7 days), the pH value of WGRC decreases. This is mainly because $Ca(OH)_2$ reacts with $CO_2$ to form $CaCO_3$, leading to a decrease in pH. It is important to note that during this stage, the pH value of low silica-calcium ratio samples decreases more rapidly. This suggests that the calcium content in these samples is consumed quickly due to carbonation, making it difficult to provide sufficient alkaline buffering capacity. As a result, the C-S-H gel structure is damaged, which allows further $CO_2$ penetration. For samples with moderate silica-calcium ratios, the pH value remains above 11.8 at 7 days, indicating that their C-S-H gel structure is dense and has a strong ability to block $CO_2$. Furthermore, for low silica-calcium ratio samples, the pH value after 28 days of carbonation decreases to 9.5~10.1, approaching the critical value for reinforcing steel corrosion, indicating weak carbonation resistance [68]. High silica-calcium ratio samples initially have a lower $Ca(OH)_2$ content, leading to an unstable C-S-H gel structure. In the later stages, severe decalcification accelerates $CO_2$ diffusion, causing the pH value to decrease rapidly.

## 6.3 Sulphate attack

The mass loss rate is an important indicator of the durability degradation of concrete under sulphate attack [69]. The calculation results of the mass loss rate for WGRC with different silica-calcium ratios are shown in Table 13. As seen from the table, the lowest mass loss rate occurs when the silica-calcium ratio is 0.52 (for example, after 180 wet-dry cycles, the mass loss rate is 0.54%). However, when the silica-calcium ratio is too low (such as 0.34) or too high (such as 0.76), the mass loss rate increases significantly, reaching 1.25% and 2.45%, respectively. This phenomenon can be explained from the perspective of the microstructure of C-S-H gel and the changes in cement hydration products. Wang et al. [70] pointed out that when the calcium and silica contents are moderate, the C-S-H gel structure becomes more dense and stable, with an appropriate amount of calcium hydroxide, which enhances the concrete's resistance to sulphate attack. Yuan et al. [71] also made a similar observation, suggesting that a balanced amount of calcium and silica helps form a stronger C-S-H gel structure, which improves the concrete's durability. In contrast, when the silica-calcium ratio is too high, the silicate phase dominates the C-S-H gel. Since this phase has weaker strength, the concrete is more prone to microcracks during the wet-dry cycles, leading to a higher mass loss rate [72]. On the other hand, when the silica-calcium ratio is too low, the amount of calcium hydroxide increases significantly. This excess calcium hydroxide can react with sulphates, producing expansive products that generate internal stress in the concrete, leading to cracking and spalling, which significantly increases the mass loss rate [73,74].

In the concrete wet-dry cycle sulphate attack experiment, ultrasonic velocity is an important indicator for evaluating internal damage and microstructural changes in concrete [75,76]. The ultrasonic velocity test results for WGRC with different silica-calcium ratios are shown in Table 14. The results show that, as the wet-dry cycle increases, the ultrasonic velocity of WGRC with different silica-calcium ratios gradually decreases. For example, when the silica-calcium

**Table 12. Experimental results of pH value.**

| Silica-calcium ratio | Carbonization time | | | |
|---|---|---|---|---|
| | 3d | 7d | 14d | 28d |
| 0.34 | 11.5 | 11.0 | 10.2 | 9.5 |
| 0.40 | 11.8 | 11.3 | 10.5 | 9.8 |
| 0.46 | 12.0 | 11.5 | 10.8 | 10.1 |
| 0.52 | 12.3 | 11.8 | 11.1 | 10.5 |
| 0.58 | 12.5 | 12.0 | 11.4 | 10.8 |
| 0.64 | 12.4 | 11.9 | 11.2 | 10.6 |
| 0.70 | 12.2 | 11.7 | 11.0 | 10.3 |
| 0.76 | 12.0 | 11.5 | 10.8 | 10.1 |

**Table 13. Calculation results of mass loss rate.**

| Silica-calcium ratio | Number of wet-dry cycles | | | | | |
|---|---|---|---|---|---|---|
| | 30 | 60 | 90 | 120 | 150 | 180 |
| 0.34 | 0.18 | 0.42 | 0.63 | 0.81 | 1.02 | 1.25 |
| 0.40 | 0.14 | 0.32 | 0.48 | 0.65 | 0.80 | 0.98 |
| 0.46 | 0.12 | 0.25 | 0.37 | 0.50 | 0.62 | 0.75 |
| 0.52 | 0.09 | 0.18 | 0.27 | 0.36 | 0.45 | 0.54 |
| 0.58 | 0.15 | 0.30 | 0.47 | 0.63 | 0.78 | 0.95 |
| 0.64 | 0.22 | 0.45 | 0.68 | 0.90 | 1.12 | 1.35 |
| 0.70 | 0.30 | 0.62 | 0.93 | 1.25 | 1.55 | 1.85 |
| 0.76 | 0.40 | 0.82 | 1.23 | 1.65 | 2.05 | 2.45 |

**Table 14. Ultrasonic wave velocity test results (Unit: km/s).**

| Silica-calcium ratio | Number of wet-dry cycles | | | | | | |
|---|---|---|---|---|---|---|---|
| | 0 | 30 | 60 | 90 | 120 | 150 | 180 |
| 0.34 | 4.50 | 4.45 | 4.38 | 4.30 | 4.22 | 4.15 | 4.08 |
| 0.40 | 4.55 | 4.52 | 4.48 | 4.42 | 4.36 | 4.30 | 4.24 |
| 0.46 | 4.60 | 4.58 | 4.55 | 4.52 | 4.48 | 4.44 | 4.36 |
| 0.52 | 4.65 | 4.64 | 4.62 | 4.60 | 4.58 | 4.56 | 4.54 |
| 0.58 | 4.58 | 4.55 | 4.52 | 4.47 | 4.43 | 4.39 | 4.34 |
| 0.64 | 4.55 | 4.50 | 4.45 | 4.40 | 4.35 | 4.30 | 4.25 |
| 0.70 | 4.50 | 4.43 | 4.36 | 4.29 | 4.22 | 4.15 | 4.08 |
| 0.76 | 4.45 | 4.38 | 4.31 | 4.24 | 4.17 | 4.10 | 4.03 |

ratio is 0.76, the ultrasonic velocity decreases from 4.38 km/s after 30 cycles to 4.03 km/s after 180 cycles. Similar results have been reported in previous studies on different types of concrete [77–81]. Genovés et al. [78] suggested that sulphate attack leads to the expansion of microcracks, increased porosity, damage to the interfacial zone, and the dissolution and decalcification of hydration products, all of which cause a continuous decrease in the ultrasonic velocity. It should be noted that when the silica-calcium ratio is 0.52, the ultrasonic velocity decreases the slowest with the increase in wet-dry cycles. For example, the initial ultrasonic velocity is 4.65 km/s, and after 180 cycles, it decreases to 4.54 km/s, a reduction of only 2.6%. This indicates that at a silica-calcium ratio of 0.52, the internal structure of the concrete is most stable, and the damage caused by sulphate attack to the microstructure is minimal. Moreover, as shown in Table 14, both excessively low and high silica-calcium ratios lead to a significant decrease in ultrasonic velocity. This is consistent with previous reports.

## 7. Conclusion

In this study, the silicon-to-calcium (Si/Ca) ratio was employed as the central regulation parameter to investigate the physical–mechanical properties, durability evolution, and microstructural mechanisms of glass sand–glass powder–rice husk ash concrete (WGRC). A shear damage constitutive model was also proposed and validated. The main conclusions are as follows:

(1) The Si/Ca ratio significantly affects WGRC performance. Within the range of 0.52–0.58, WGRC exhibits optimal comprehensive strength, impermeability, and durability. Both excessively low and high Si/Ca ratios inhibit the formation of dense C-S-H gels, resulting in porous structures and degraded performance.

(2) With increasing curing age, the Si/Ca ratio of the C-S-H gel increases gradually, and the microstructure evolves from flocculent to layered forms. Samples with moderate Si/Ca ratios show a strong synergistic effect between C-S-H and AFt, improving matrix compactness and interfacial structure.

(3) Shear test results show that moderate Si/Ca ratio concrete exhibits the highest shear strength at a 25° shear angle. The continuum damage mechanics–based model accurately captures pre-peak linearity, post-peak softening, and brittleness, demonstrating strong adaptability to varying mix proportions and loading paths.

(4) Low Si/Ca ratios favor the formation of abundant C-S-H gels and dense matrices. Moderate ratios lead to synergistic growth of C-S-H and AFt, forming optimal microstructures. At high Si/Ca ratios, limited calcium supply causes discontinuous gel structures and increased porosity, weakening microstructural integrity.

(5) WGRC with an optimal Si/Ca ratio shows the lowest mass loss rate, electric flux, carbonation depth, and pH decline rate, confirming enhanced resistance to sulfate attack and carbonation. These findings validate the microstructural role of Si/Ca regulation in durability improvement.

However, the study does not yet fully address the system-level impact of Si/Ca ratio regulation on WGRC performance. Future work should explore its interaction with environmental factors such as humidity, temperature, and alkaline activation. Introducing other solid waste types and applying data-driven approaches (e.g., machine learning) may further refine mixture design and predictive modeling, supporting broader application of WGRC in aggressive environments. It should be noted that the present microstructural interpretation is primarily based on EDS-derived gel composition and SEM morphology; therefore, the discussion is semi-quantitative in nature. More advanced quantitative techniques such as XRD Rietveld refinement, TGA, or MIP may be employed in future work to further verify phase evolution and pore structure changes.

## Author contributions

**Conceptualization:** Jiayuan Lou, yiwen liang, Wenhua Zha.

**Data curation:** yiwen liang, Wenhua Zha, Qiang Su.

**Resources:** Wenhua Zha.

**Writing – original draft:** Jiayuan Lou, yiwen liang, Qiang Su.

**Writing – review & editing:** yiwen liang, Qiang Su.

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
