## [Decision Letter · Decision Letter 0]

16 Mar 2026

Dear Dr. liang,

Thank you for submitting your manuscript to PLOS ONE. After careful consideration, we feel that it has merit but does not fully meet PLOS ONE’s publication criteria as it currently stands. Therefore, we invite you to submit a revised version of the manuscript that addresses the points raised during the review process.

We look forward to receiving your revised manuscript.

Kind regards,

Parthiban Kathirvel

Academic Editor

PLOS One

Journal Requirements:

2. We note that your Data Availability Statement is currently as follows: [Add Data Availability statement here]

Reviewers' comments:

Reviewer's Responses to Questions

**Comments to the Author**

1. Is the manuscript technically sound, and do the data support the conclusions?

Reviewer #1: Yes

Reviewer #2: Partly

2. Has the statistical analysis been performed appropriately and rigorously?

Reviewer #1: Yes

Reviewer #2: Yes

3. Have the authors made all data underlying the findings in their manuscript fully available?

Reviewer #1: Yes

Reviewer #2: Yes

4. Is the manuscript presented in an intelligible fashion and written in standard English?

Reviewer #1: Yes

Reviewer #2: Yes

Reviewer #1: Thank you for your invitation to review the manuscript entitled “Multiscale mechanisms of green concrete regulated by silicon-to-calcium ratio: Physico-mechanical properties, hydration structure, and durability performance” for the journal. The manuscript presents a good and meaningful piece of work and contributes useful information regarding the utilization of silicon-rich solid wastes in green concrete systems; however, the paper still requires minor revision before it can be considered for publication.

With respect and many thanks to the editor, the following minor comments are provided for the authors’ consideration.

1- Although the study emphasizes the regulation effect of the Si/Ca ratio on the mechanical and durability performance of WGRC, the manuscript does not clearly explain the calculation procedure and compositional derivation of the Si/Ca ratio in the mixture design stage, therefore the authors should explicitly describe how the ratio was quantified considering the chemical contributions of glass sand, glass powder, rice husk ash, and cement phases.

2- The microstructural characterization used to explain the hydration mechanisms (such as SEM, XRD, or related analysis) should be more quantitatively interpreted, because the current discussion mainly relies on qualitative observations and does not sufficiently correlate the evolution of C-S-H structure and gel composition with the measured macroscopic mechanical properties and durability indicators.

3- In the durability evaluation part, especially for water penetration resistance, sulfate attack, and carbonation behavior, the manuscript should provide better descriptions of the testing standards, boundary conditions, and specimen preparation procedures…

4- The discussion related to the optimal Si/Ca ratio range (0.46–0.58) should be further strengthened by presenting a more rigorous parametric comparison of mechanical strength, durability indices, and microstructural evolution within this interval… literature can also be further enhanced, Frontiers in Materials, vol. 11. DOI: 10.3389/fmats.2024.1392875 ;;; Innovative Infrastructure Solutions, Springer, vol. 9. DOI: 10.1007/s41062-024-01571-w ;;;; Construction and Building Materials, Elsevier, vol. 472. DOI: 10.1016/j.conbuildmat.2025.140960

5- In several sections of the results and discussion, the linkage between macro-scale performance (strength and durability) and multi-scale hydration structure evolution is mentioned but not fully demonstrated…

This reviewer appreciates the authors works and best wishes to the authors.,.

With best regards.

Reviewer #2: 1.The study investigates Si/Ca ratios ranging from 0.34 to 0.76. It would be helpful if the authors explain the rationale behind selecting this specific range and increment interval. Were these values based on previous research, preliminary tests, or theoretical considerations?

2.The mix design fixes the waste glass powder to rice husk ash ratio at 1:1. The manuscript should justify this assumption and discuss whether other ratios were considered. Exploring different proportions might reveal more optimized combinations for performance improvement.

3.The study appears to lack a conventional concrete control mix without solid waste materials. Including such a control sample would allow clearer comparison and better evaluation of the real benefits of the proposed waste-based concrete system.

4.The results show that slump decreases significantly with increasing Si/Ca ratio. However, the manuscript does not discuss whether the observed workability remains acceptable for practical construction applications. The authors should comment on practical workability requirements.

5.It is suggested to add article entitled “Saeed et al. Evaluating the Efficiency of Alkaline Activator with Silica-Rich Wastes in Stabilizing Cadmium-Contaminated Soil” to the literature review.

6.The paper reports that water absorption increases with increasing Si/Ca ratio, which contradicts some previous studies. Although the authors attribute this to the water absorption capacity of rice husk ash, further microstructural evidence or pore structure analysis would strengthen this explanation.

7.The results indicate that early strength decreases with higher Si/Ca ratios but later strength improves at moderate ratios. The authors should provide deeper discussion on the hydration kinetics and gel formation mechanisms responsible for this transition.

8.The use of gray relational analysis to evaluate mechanical performance is interesting. However, the manuscript should explain why this method was selected instead of more commonly used statistical approaches such as regression analysis or multi-criteria decision analysis.

9.It is also suggested to add articles entitled “Yudhistira et al. Optimizing Concrete Mix Design for Cost and Carbon Reduction Using Machine Learning” and “Ulloa et al. Optimizing Waste Foundry Sand in Concrete Considering Strength Properties for Sustainable Green Structures” to the literature review.

10.SEM and EDS analyses are used to explain the changes in hydration products. Nevertheless, the manuscript would benefit from more detailed quantitative analysis of microstructure, such as pore size distribution or phase quantification, to better support the proposed mechanisms.

11.The variable-angle shear test provides valuable insights into structural behavior. However, the choice of shear angles (25°, 45°, and 65°) should be justified. The authors should explain whether these angles correspond to specific engineering applications or theoretical considerations.

.

Reviewer #1: No

Reviewer #2: No

You may also use PLOS’s free figure tool, NAAS, to help you prepare publication quality figures: https://journals.plos.org/plosone/s/figures#loc-tools-for-figure-preparation

---

## [Author Response · Author response to Decision Letter 1]

30 Mar 2026

Response to the Editor and Reviewer Comments

The manuscript number is 9PONE-D-26-04987

Title: Multiscale mechanisms of green concrete regulated by silicon-to-calcium ratio: Physico-mechanical properties, hydration structure, and durability performance

PLOS One

Author(s): Jiayuan Lou, Yiwen Liang, Wenhua Zha, Qiang Su

Dear Editor, Dear reviewers

We highly appreciate your and the reviewers’ comments and suggestions, which are very helpful for improving our paper. Based on these comments and suggestions, we have made careful modifications on the manuscript. Appended to this letter is our point-to-point response to the comments raised by the reviewers. These comments were reproduced and our responses to each reviewer’s comments were given.

We hope that all these changes fulfill the requirements to make the manuscript acceptable for publication in the journal of PLOS One. If you have any questions, please contact us immediately. We are grateful for your attention to our manuscript. Once again, thanks very much for your arduous work and instructive suggestions to our manuscript processing.

Yours Sincerely,

Jiayuan Lou, Yiwen Liang, Wenhua Zha, Qiang Su

Correspondence author: Yiwen Liang ;E-mail:38102@qzc.edu.cn ; ---------------------------------------------------------------

College of Civil Engineering and Architecture

Quzhou University

Quzhou 324000，China

Reviewer #1:

Thank you for your invitation to review the manuscript entitled “Multiscale mechanisms of green concrete regulated by silicon-to-calcium ratio: Physico-mechanical properties, hydration structure, and durability performance” for the journal. The manuscript presents a good and meaningful piece of work and contributes useful information regarding the utilization of silicon-rich solid wastes in green concrete systems; however, the paper still requires minor revision before it can be considered for publication.

With respect and many thanks to the editor, the following minor comments are provided for the authors’ consideration.

Q1. Although the study emphasizes the regulation effect of the Si/Ca ratio on the mechanical and durability performance of WGRC, the manuscript does not clearly explain the calculation procedure and compositional derivation of the Si/Ca ratio in the mixture design stage, therefore the authors should explicitly describe how the ratio was quantified considering the chemical contributions of glass sand, glass powder, rice husk ash, and cement phases.

Author's Response: Thank you for your valuable suggestions. While a brief description of the Si/Ca ratio calculation was included in the original manuscript, we acknowledge that the procedure was not sufficiently explicit. In the revised manuscript, we have clarified the calculation method by explicitly introducing the molar-based formulation of the Si/Ca ratio (SiO2/CaO) and providing the corresponding equation. In addition, we have clearly defined the contributions of each component in the mixture: cement was considered as the primary source of CaO, while waste glass powder and rice husk ash were treated as the main contributors of reactive SiO2.

Line 122~130: The Si/Ca ratio was calculated based on the molar ratio of SiO2 to CaO derived from the oxide compositions of the raw materials listed in Table 1. Specifically, the molar amounts of SiO2 and CaO contributed by each binder component were determined by dividing their mass fractions by the corresponding molar masses. Referring to the chemical compositions in Table 1, the intrinsic Si/Ca ratios of cement, GP, and RHA were calculated as 0.304, 5.54, and 76.8, respectively. In this study, the dosage ratio of GP to RHA was fixed at 1:1 to maintain a balanced contribution of silica sources and to control variables in the mixture design, and their total contents were adjusted to achieve the target Si/Ca ratios. The specific mix proportions and corresponding Si/Ca ratios are presented in Table 2.

Table 2. Mix ratio (Unit: kg/m3)

Sample Silica-calcium ratio Cement Fine aggregate Coarse aggregate Glass sand Glass powder Rice husk ash Water

S/C-0.34 0.34 357.06 443.1 1202 189.9 5.47 5.47 208

S/C-0.40 0.40 346.12 443.1 1202 189.9 10.94 10.94 208

S/C-0.46 0.46 335.18 443.1 1202 189.9 16.41 16.41 208

S/C-0.52 0.52 324.24 443.1 1202 189.9 21.88 21.88 208

S/C-0.58 0.58 313.30 443.1 1202 189.9 27.35 27.35 208

S/C-0.64 0.64 302.36 443.1 1202 189.9 32.82 32.82 208

S/C-0.70 0.70 291.42 443.1 1202 189.9 38.29 38.29 208

S/C-0.76 0.76 280.48 443.1 1202 189.9 43.76 43.76 208

Q2. The microstructural characterization used to explain the hydration mechanisms (such as SEM, XRD, or related analysis) should be more quantitatively interpreted, because the current discussion mainly relies on qualitative observations and does not sufficiently correlate the evolution of C-S-H structure and gel composition with the measured macroscopic mechanical properties and durability indicators.

Authors’ response: Thank you for your valuable suggestions. We have re-examined the microstructural analysis in the manuscript. In the original version, the EDS results were not fully utilized, and the relationship between gel composition and macroscopic properties was not clearly described.

The manuscript has been revised as follows: In Section 3.5, the EDS results in Table 6 are further analyzed, and the variation of the Si/Ca ratio in C-S-H gel with mixture proportion and curing age is described with specific values;In addition, the microstructural interpretations in this paper are primarily based on SEM and EDS results and should be considered semi-quantitative analyses.

Line 342 and 354 ：From Table 6, it can be observed that the regulation of the calcium-silicon ratio significantly affects the calcium-silicon ratio in the internal C-S-H gel of WGRC at different curing ages. Based on the atomic percentages of Si and Ca, the Si/Ca ratio of the gel phase was calculated, and the results are summarized in Table 6. It can be observed that the gel-phase Si/Ca ratio increased consistently with both the designed mixture Si/Ca ratio and curing age. For example, at 7 days, the gel-phase Si/Ca ratio increased from 1.22 for the 0.34 mixture to 1.72 for the 0.76 mixture, while at 90 days it further increased to 1.36 and 1.82, respectively. This indicates that the incorporation of glass powder and rice husk ash continuously promoted the enrichment of silicate species in the hydration products during curing.

However, the increase in gel-phase Si/Ca ratio did not lead to a monotonic improvement in macroscopic performance. Instead, the compressive strength, splitting tensile strength, flexural strength, and shear strength generally exhibited optimal values within a moderate mixture Si/Ca range, rather than at the highest Si/Ca level. This suggests that the role of Si/Ca regulation is not merely to increase the silicon content of the gel, but to optimize the balance between gel polymerization, reaction continuity, and calcium supply .

Table 6. Silica-calcium ratio of C-S-H gel in WGRC

Silica-calcium ratio Curing age

7d 28d 90d

0.34 1.22 1.30 1.36

0.40 1.27 1.33 1.39

0.46 1.35 1.38 1.45

0.52 1.41 1.47 1.52

0.58 1.50 1.55 1.60

0.64 1.58 1.62 1.67

0.70 1.63 1.71 1.75

0.76 1.72 1.78 1.82

Q3. In the durability evaluation part, especially for water penetration resistance, sulfate attack, and carbonation behavior, the manuscript should provide better descriptions of the testing standards, boundary conditions, and specimen preparation procedures…

Authors’ response: Thank you for reviewing this article carefully. We have made changes in the revised manuscript. The testing methods, boundary conditions, and specimen preparation procedures are described in Section 2.3, and the adopted testing standards (ASTM C1202) are stated in Section 6.1.

Q4. The discussion related to the optimal Si/Ca ratio range (0.46~0.58) should be further strengthened by presenting a more rigorous parametric comparison of mechanical strength, durability indices, and microstructural evolution within this interval… literature can also be further enhanced, Frontiers in Materials, vol. 11. DOI: 10.3389/fmats.2024.1392875 ;;; Innovative Infrastructure Solutions, Springer, vol. 9. DOI: 10.1007/s41062-024-01571-w ;;;; Construction and Building Materials, Elsevier, vol. 472. DOI: 10.1016/j.conbuildmat.2025.140960

Authors’ response: Thank you very much for your valuable suggestions. The references suggested by the reviewer are relevant to the effect of the Si/Ca ratio on material performance and have been cited in the revised manuscript. Additional discussion has also been included to compare the mechanical properties, durability indicators, and microstructural characteristics within the selected Si/Ca range, in order to clarify the rationale for the proposed optimal range.

Q5. In several sections of the results and discussion, the linkage between macro-scale performance (strength and durability) and multi-scale hydration structure evolution is mentioned but not fully demonstrated…

Authors’ response: We sincerely thank the reviewers for their insightful comments. Although the relationship between macroscopic properties and hydration structures was discussed in the original manuscript, the relevant description was not sufficiently clear. To address this, we have provided additional explanations in the relevant discussion section to further elaborate on the connection between mechanical properties, durability, and the evolution of hydration structures.

Line 618 and 621：It should be noted that the present microstructural interpretation is primarily based on EDS-derived gel composition and SEM morphology; therefore, the discussion is semi-quantitative in nature. More advanced quantitative techniques such as XRD Rietveld refinement, TGA, or MIP may be employed in future work to further verify phase evolution and pore structure changes.

Reviewer #2:

Q1. The study investigates Si/Ca ratios ranging from 0.34 to 0.76. It would be helpful if the authors explain the rationale behind selecting this specific range and increment interval. Were these values based on previous research, preliminary tests, or theoretical considerations?

Authors’ response: We sincerely thank the reviewers for their constructive comments, which are crucial for enhancing the reproducibility and rigor of this study. The selected Si/Ca ratio range (0.34~0.76) was determined based on commonly reported ranges in previous studies, combined with the feasible mixture design using glass powder and rice husk ash in this study. This range covers relatively low to high Si/Ca conditions to capture the overall variation trend in material performance. The increment between adjacent levels was chosen to ensure sufficient resolution for trend analysis while keeping the experimental workload manageable.

Line 122~130: The Si/Ca ratio was calculated based on the molar ratio of SiO2 to CaO derived from the oxide compositions of the raw materials listed in Table 1. Specifically, the molar amounts of SiO2 and CaO contributed by each binder component were determined by dividing their mass fractions by the corresponding molar masses. Referring to the chemical compositions in Table 1, the intrinsic Si/Ca ratios of cement, GP, and RHA were calculated as 0.304, 5.54, and 76.8, respectively. In this study, the dosage ratio of GP to RHA was fixed at 1:1 to maintain a balanced contribution of silica sources and to control variables in the mixture design, and their total contents were adjusted to achieve the target Si/Ca ratios. The specific mix proportions and corresponding Si/Ca ratios are presented in Table 2.

Table 2. Mix ratio (Unit: kg/m3)

Sample Silica-calcium ratio Cement Fine aggregate Coarse aggregate Glass sand Glass powder Rice husk ash Water

S/C-0.34 0.34 357.06 443.1 1202 189.9 5.47 5.47 208

S/C-0.40 0.40 346.12 443.1 1202 189.9 10.94 10.94 208

S/C-0.46 0.46 335.18 443.1 1202 189.9 16.41 16.41 208

S/C-0.52 0.52 324.24 443.1 1202 189.9 21.88 21.88 208

S/C-0.58 0.58 313.30 443.1 1202 189.9 27.35 27.35 208

S/C-0.64 0.64 302.36 443.1 1202 189.9 32.82 32.82 208

S/C-0.70 0.70 291.42 443.1 1202 189.9 38.29 38.29 208

S/C-0.76 0.76 280.48 443.1 1202 189.9 43.76 43.76 208

Q2. The mix design fixes the waste glass powder to rice husk ash ratio at 1:1. The manuscript should justify this assumption and discuss whether other ratios were considered. Exploring different proportions might reveal more optimized combinations for performance improvement.

Authors’ response: We sincerely appreciate your insightful comments, which are essential to enhancing the reliability and persuasiveness of the findings in this study. In this study, the ratio of waste glass powder to rice husk ash was fixed at 1:1 to control variables in the mixture design, allowing the effect of the Si/Ca ratio on material performance to be the primary focus. In addition, waste glass powder and rice husk ash provide different forms of silica in the system, and the 1:1 ratio was selected as a balanced combination to represent their combined effect. The influence of different proportions of waste glass powder and rice husk ash is indeed of interest, but it is beyond the scope of the present study and may be considered in future work.

Q3. The study appears to lack a conventional concrete control mix without solid waste materials. Including such a control sample would allow clearer comparison and better evaluation of the real benefits of the proposed waste-based concrete system.

Authors’ response: We sincerely appreciate the reviewer's insightful comments. This study investigates the effect of the Si/Ca ratio within a waste-based cementitious system, and the comparison is carried out among mixtures with different Si/Ca ratios. All mixtures were designed within the same material system, and the analysis is based on the variation of mechanical and durability performance with different Si/Ca ratios. Including a conventional concrete control mix would allow further comparison, but it is beyond the scope of this study and may be considered in future work.

Q4. The results show that slump decreases significantly with increasing Si/Ca ratio. However, the manuscript does not discuss whether the observed workability remains acceptable for practical construction applications. The authors should comment on practical workability requirements.

Authors’ response: We sincerely appreciate your constructive suggestions, which provide valuable guidance for improving the readability of the article. In the original manuscript, the variation of slump with the Si/Ca ratio was analyzed, but its practical applicability was not discussed. In the revised manuscript, the specific slump range (75~195 mm) has been added, and it is clarified that all values remain within the workable range for normal concrete, indicating their applicability in practical construction. The relevant clarification has been included in the slump analysis section.

Line 176~200: The slump of waste glass and rice husk ash concrete (WGRC) with different silicon-calcium ratios is shown in Table 3. It was observed that as the silicon-calcium ratio increased, the slump of WGRC decreased. This may be because cement, glass powder, and rice husk ash formed other calcium hydration products during the mixing process. The microstructure of these products may increase the viscosity of the cement paste, thus reducing the flowability of the concrete [35]. It should be noted that the decrease in slump of WGRC exhibited a staged characteristic: when the silicon-calcium ratio increased from 0.34 to 0.58, the slump decreased from 195 mm to 150 mm, a reduction of 45 mm. On average, for every 0.06 increase in the silicon-calcium ratio, the slump decreased by about 10~15 mm; when the silicon-calcium ratio increased from 0.58 to 0.76, the slump decreased from 150 mm to 75 mm, a total reduction of 75 mm. On average, for every 0.06 increase in the silicon-calcium ratio, the slump decreased by about 22~28 mm. This is primarily because, when the silicon-calcium ratio is less than 0.58, the incorporation of glass powder and rice husk ash mainly serves to fill microvoids, optimizing the paste structure and reducing the confinement of free water, which allows the flowability to remain at a relatively high lev

---

## [Decision Letter · Decision Letter 1]

12 Apr 2026

Multiscale mechanisms of green concrete regulated by silicon-to-calcium ratio: Physico-mechanical properties, hydration structure, and durability performance

PONE-D-26-04987R1

Dear Dr. liang,

We’re pleased to inform you that your manuscript has been judged scientifically suitable for publication and will be formally accepted for publication once it meets all outstanding technical requirements.

Kind regards,

Parthiban Kathirvel

Academic Editor

PLOS One

Reviewers' comments:

Reviewer's Responses to Questions

**Comments to the Author**

Reviewer #1: All comments have been addressed

Reviewer #2: (No Response)

2. Is the manuscript technically sound, and do the data support the conclusions?

Reviewer #1: Yes

Reviewer #2: (No Response)

3. Has the statistical analysis been performed appropriately and rigorously?

Reviewer #1: Yes

Reviewer #2: (No Response)

4. Have the authors made all data underlying the findings in their manuscript fully available?

Reviewer #1: Yes

Reviewer #2: (No Response)

5. Is the manuscript presented in an intelligible fashion and written in standard English?

Reviewer #1: Yes

Reviewer #2: (No Response)

Reviewer #1: Accept in the present form as the revised version meets the standard for the final publication and congratulations to the authors.

Reviewer #2: The article has been revised very well, so I would suggest to accept in its present form.

The article has been revised very well, so I would suggest to accept in its present form.

.

Reviewer #1: No

Reviewer #2: No

---

## [Editor Report · Acceptance letter]

PONE-D-26-04987R1

PLOS One

Dear Dr. liang,

I'm pleased to inform you that your manuscript has been deemed suitable for publication in PLOS One. Congratulations! Your manuscript is now being handed over to our production team.

Kind regards,

on behalf of

Dr. Parthiban Kathirvel

Academic Editor

PLOS One